# Capacitive tendency concept alongside supervised machine-learning toward classifying electrochemical behavior of battery and pseudocapacitor materials

Siraprapha Deebansok[1], Jie Deng[2], Etienne Le Calvez[3,4], Yachao Zhu [5], Olivier Crosnier[3,4], Thierry Brousse [3,4] & Olivier Fontaine [1,6] ✉

In recent decades, more than 100,000 scientific articles have been devoted to the development of electrode materials for supercapacitors and batteries. However, there is still intense debate surrounding the criteria for determining the electrochemical behavior involved in Faradaic reactions, as the issue is often complicated by the electrochemical signals produced by various electrode materials and their different physicochemical properties. The difficulty lies in the inability to determine which electrode type (battery vs. pseudocapacitor) these materials belong to via simple binary classification. To overcome this difficulty, we apply supervised machine learning for image classification to electrochemical shape analysis (over 5500 Cyclic Voltammetry curves and 2900 Galvanostatic Charge-Discharge curves), with the predicted confidence percentage reflecting the shape trend of the curve and thus defined as a manufacturer. It's called "capacitive tendency". This predictor not only transcends the limitations of human-based classification but also provides statistical trends regarding electrochemical behavior. Of note, and of particular importance to the electrochemical energy storage community, which publishes over a hundred articles per week, we have created an online tool to easily categorize their data.

In the energy storage research field, batteries are one of the most studied types of devices owing to their use in a wide range of applications including electronic equipment, electric vehicles and for medical and military purposes[1]. On the other hand, pseudocapacitive electrodes have attracted a considerable amount of attention due to their superior power capability[2]. Both of these energy storage systems are generally composed of various types of electrode materials exhibiting electrochemical signals that may or may not resemble one another[3].

It is common knowledge that electric double layer capacitors (EDLCs) rely on a non-faradaic process without any electron transfer, whereas batteries and pseudocapacitors are governed by faradaic reactions[4]. The latter processes are generally depicted by peaks on Cyclic Voltammograms (CVs) and plateaus on Galvanostatic Charge-Discharge (GCD) curves (Fig. 1)[5]. However, some faradaic electrode materials including pseudocapacitors display electrochemical signals similar to those of EDLCs, such as the rectangular/quasi-rectangular CV

[1]Molecular Electrochemistry for Energy laboratory, VISTEC, Institute of Science and Technology, Rayong 21210, Thailand. [2]Institute for Advanced Study & College of Food and Biological Engineering, Chengdu University, Chengdu 610106, China. [3]Nantes Université, CNRS, Institut des Matériaux de Nantes Jean Rouxel, IMN, 44000 Nantes, France. [4]Réseau sur le Stockage Électrochimique de l'Énergie (RS2E), CNRS FR 3459, 33 rue Saint Leu, 80039 Amiens, France. [5]ICGM, Université de Montpellier, CNRS, 34293 Montpellier, France. [6]Institut Universitaire de France, 75005 Paris, France. ✉e-mail: olivier.fontaine@vistec.ac.th

**Fig. 1 | Classification of CVs and GCDs curves.** Illustration of **a** experimental CVs and GCDs of different electrode materials including MnO$_2$[39], V$_2$C[40], RuO$_x$[41], LaMnO$_3$[42], Ti$_3$C$_2$T$_x$[15], H$_2$TiNb$_6$O$_{18}$[43], Ag$_{1-3x}$La$_x\square_{2x}$NbO$_3$[44], Nb$_2$O$_5$[45], nano-MnS$_2$[46], bulk-MoS$_2$[46], TiO$_2$[47] and NaFePO$_4$[48], theoretical **b** CVs and **c** GCDs undergoing different electrochemical processes.

and the sloping GCD curves[6,7], found in a variety of transition metal oxides (RuO$_2$[8], MnO$_2$[9,10]), conducting polymers (poly(3,4-ethylenedioxythiophene)[11,12], polyaniline[13,14]), and carbides (MXene)[15]. Currently, owing to the vast amounts of materials studied, guidelines for distinguishing between the two are still largely inadequate, with some studies even contradicting the conventional definition of Conway et al., as later supported by Brousse et al. and other researchers in the field[7].

Indeed, electrochemical signals are numerous and complex, varying according to the choice of electrode materials, as shown in Fig. 1, hence the difficulty in identifying and categorizing these materials based on electrochemical signals. Recently, Fleischmann et al.[16], emphasized on the importance of a unified understanding when it comes to the electrochemical signals found in supercapacitors and batteries. The authors proposed the concept of the 'continuum transition' where the overlapping of electrochemical signals (between battery and supercapacitor) lies in this region depending on the electrolyte confinement stage. It signifies that understanding this overlapping transition essentially requires a clear-cut classification of electrode material types based on their electrochemical behaviors (in CV and GCD). Unfortunately, this qualitative concept of 'continuum spectrum' is urgently required an informative transformation to obtain the quantitative value of it. In order to complete this concept of 'continuum spectrum' and to provide the real quantitative value to it, we analyze the electrochemical signals with the help of supervised machine-learning for achieving the descriptor, "capacitive tendency" that allows our community to quantify this important spectrum.

To date, computing techniques have been used as somewhat satisfactory tools toward ascertaining the charge storage mechanism behind various electrochemical signatures[17–20]. Recently, text-mining algorithms have been developed to efficiently extract various specific information of the materials from the article such as BatteryDataExtractor using bidirectional-encoder representations from transformers (BERT)[21], and Li-ion battery annotated corpus (LIBAC) based on Machine Learning (ML), natural language processing (NLP), Named Entity Recognition (NER)[22–24]. However, the direct interpretation of the image data from figures remain difficult using the above method of data-mining from image. Machine learning has been used to predict the electrochemical mechanism involved in the reaction that expresses through a cyclic voltammogram (CV). Deep learning has also been used to distinguish the mechanism of the electrochemical reaction from CV based on residual neural network (ResNet) architecture and

focused on analytical or fundamental electrochemistry. However, the application of machine learning to analyze electrochemical signals in the field of energy storage is still in its early stages.

This study presents that electrochemical signal analysis (CV and GCD) has been performed using a machine learning (ML) approach based on image classification. This approach is well-suited for unlabeled data, noise-tolerant, and capable of handling complex data. Ultimately, this led to the determination of the capacitive behavior of electrode materials from thousands of scientific papers. The crux of this work lies in its use of machine learning (ML) to quickly and accurately interpret electrochemical signal images and transform them into accurate values. This is made possible by the large database of electrochemical energy storage images that is available to the ML model. This approach overcomes the limitations of human ability to interpret data, which can be too complicated in most cases (Fig. 1). So, by this approach, we propose the definition call "capacitive tendency" based on the percentage confidence of the classification between box shaped and peak shaped CV, implying the capacitive behavior of electrode materials. In addition to this, we provide an online tool kit which uses supervised machine-learning to easily classify materials. Our work thus serves to put forward a new concept toward understanding and labeling the various electrochemical signatures of energy storage devices.

Image recognition is used in many fields, such as facial recognition, cancer detection and autonomous cars. All these models have been trained using a supervised or semi-supervised deep learning approach, in order to teach the model, the pattern best suited to the situation. The difference between the techniques lies in the choice of neural network, which must be adapted to the specific problem. In our case, the main difficulty was to differentiate the figures representing a CV and a GCD from the other graphs.

Overview of our study is shown in Fig. 2. In this study, these CVs and GCDs were analyzed via supervised ML trained with datasets extracted from over 4000 scientific papers. In the following section, various Convolutional Neural Network architectures are validated and selected based on the evaluations explained in the experimental section, by applying the theoretical CV and GCD curves.

Although the application of machine learning in scientific research was not uncommon before, the analysis of the shape of electrochemical signals has never appeared before. For example, Puthongkham et al. wrote a mini-review summarizing the latest applications of machine learning and experimental design in

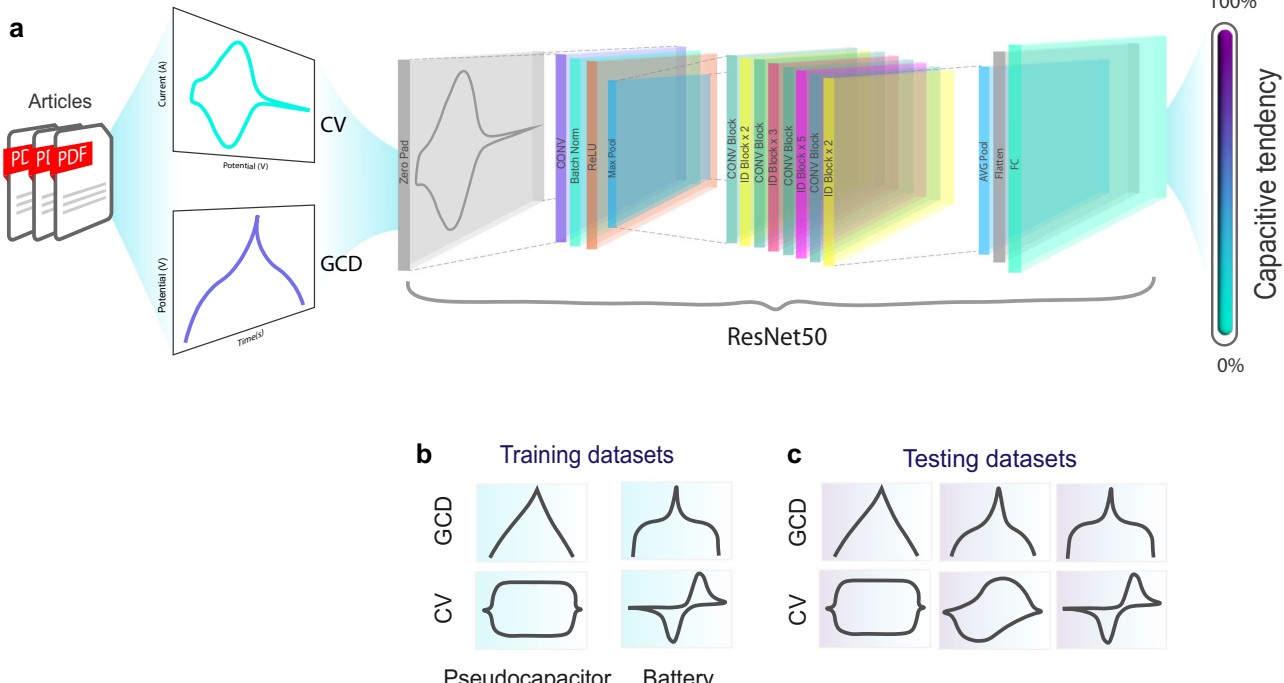

**Fig. 2 | Overview of artificial neural network training models.** Illustration of **a** Image extraction from scientific papers followed by CV and GCD classifications based on ResNet50 architecture, **b** representatives of training datasets, and **c** representative of testing datasets.

electroanalytical chemistry[25]. Khosravinia et al. used machine learning to select the best precursor to predict the specific capacitance[26]. But none of these papers focused on shape changes in electrochemical signals. The innovation of this work is to explore the shape changes between curves with the assistance of artificial intelligence, so as to find the change rules between electrochemical signals.

In comparison to other studies, the capacitive tendency analyses the shape of the electrochemical signal. Unfortunately, the capacitive tendency doesn't provide the surface contribution or the diffusional contribution inside the cyclic voltammetry.

## Dataset construction

In the present paper, all datasets are in the form of images extracted using PyMuPDF library in Python language from >3300 scientific papers. The first dataset, or Output 1, was obtained by figures extracting using OpenCV which provides (2979) GCD, (5598) CV and other images such as crystal structure image (which will not be used in the further classification steps). In the training process of GCD (process 2) and CV (process 3) classification, CV and GCD images were firstly labeled as belonging to one of two classes, namely battery or pseudocapacitor following the criteria of non-ambiguous signal shape (which can be put into four categories: (1) Box shaped CV, (2) Peak shaped CV, (3) Triangular GCD, and (4) Plateau GCD) for 80% of total data, where 20% of total data was used as testing data. These training processes is based on binary classification of electrochemical signal, such as the box vs peak shaped CV, and the triangular vs plateau shaped GCD, as represented in Supplementary Fig. 6, where all image datasets used are available on Github[27].

From Process 3, Output 3 was obtained and categorized into three types of training sets: 100% battery, 50% battery/pseudocapacitor, and 100% pseudocapacitor. This output was then further refined in Processes 4 and 5, as illustrated in Fig. 3b. We used three data sources for their study of CV and GCD images. The first source was a large dataset of over 5500 CVs and 2900 GCDs extracted from scientific papers. The second source was theoretical CVs and GCDs generated using electrochemical equations. The third source was experimental CVs and

GCDs from co-authors. Moreover, cross-validation was performed with the experts in the field to generate the different training datasets for the optimizing of the classification performance.

However, text-mining was not used in this present study since we would like to propose the simple alternative tool focusing image classification of electrochemical signals.

## Validation of classification architectures

In this work, Convolutional Neural Networks (CNNs) were selected for use as the image classification architectures[28]. Benchmarking was conducted on five different CNN models, including ResNet50[29], MobileNetV2[30], VGG16[31], Xception[32] and 8-Layer CNN[28] (see Supplementary Figs. 1, 2), to compare model performance. It was carried out based on five metrics, including: Accuracy, Sensitivity, Specificity, Precision, and F-Score[33] (see Supplementary Fig. 3 and Supplementary Eqs. 1–5). During the model training cycles, the number of training and validation iterations can impact the accuracy of the prediction since this is related to the experience gained over time by the ML model. Moreover, binary cross entropy (BCE) loss[34], calculated from the prediction error as shown in Eq. 1, was minimized along the number of training iterations to optimize predictor performance.

$$L_{BCE} = -\frac{1}{n}\left(\sum_{i=1}^{n} y_i \cdot \log(\hat{y_i}) y_i \cdot \log(\hat{y_i}) + (1 - y_i) \cdot \log(1 - \hat{y_i})\right) \quad (1)$$

Where $y_i$ is the ground truth label (0 or 1, in this case battery or pseudocapacitor), $\hat{y}$ is the predicted value, and n is the output size[34].

## Machine-learning for CV/GCD classification procedures

The ML architecture displaying the best performance after the validation step (further explained in the Results and Discussion section) was selected for use in this work as will be supervised during classification processes. ResNet50 was exploited in different steps denoted as Processes 1, 2, 3, 4, and 5 (as summarized in Fig. 3c) according to the types of inputs and outputs. All the images extracted from scientific

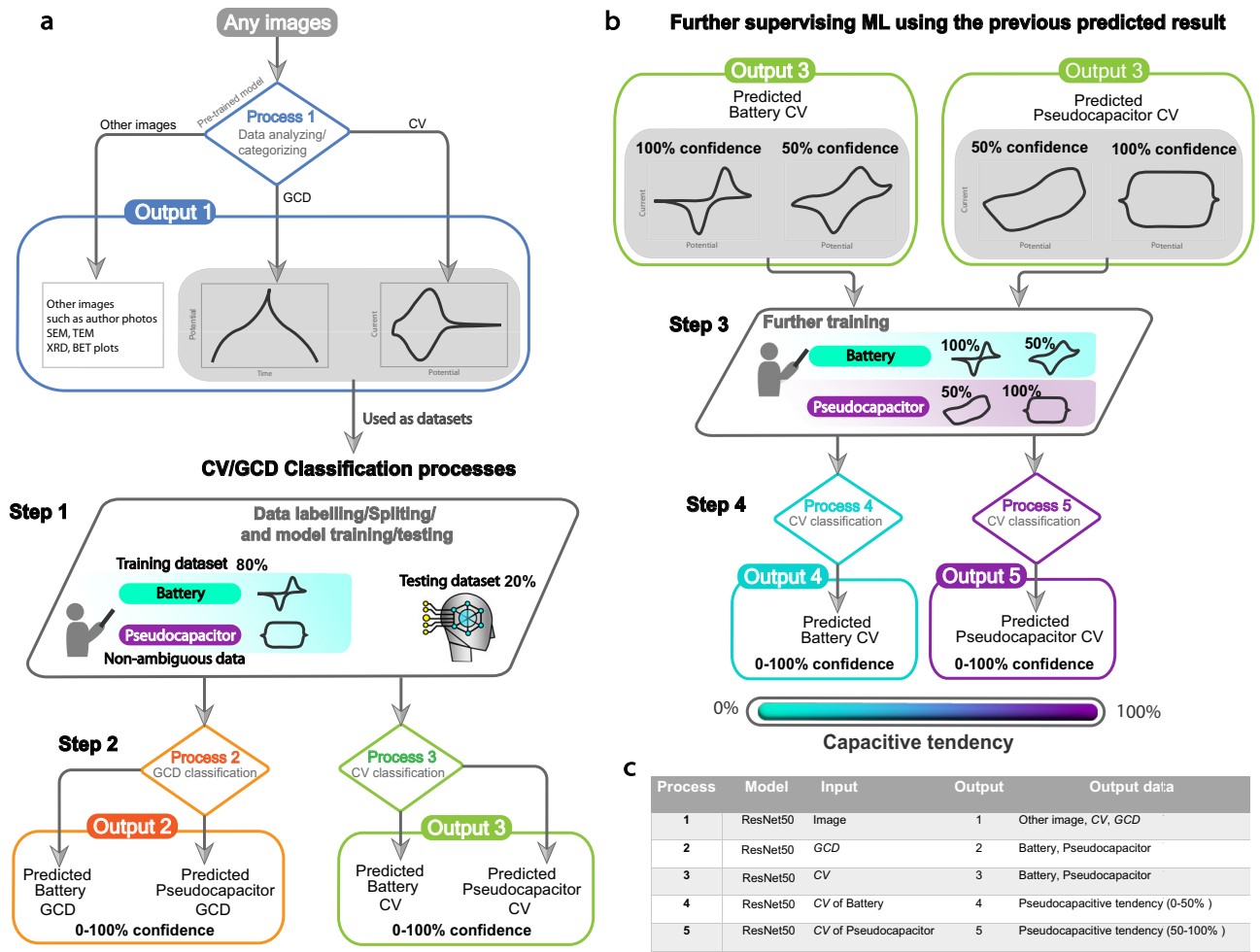

**Fig. 3 | Image classification, model training and result prediction in this work.**
**a** CV and GCD datasets obtained after classification by Process 1, splitting them into training and testing datasets for further GCD and CV classification in Process 2 and Process 3, respectively. **b** The outputs from Process 3 are used in this final classification step (process 4 and 5) to obtain the capacitive tendency based on percentage confidence rating of the prediction. **c** Table of processes, inputs and outputs performed/used to obtain these results.

papers were then categorized by Process 1 (ResNet50 model) which yielded Output 1, comprising GCDs, CVs and other images (such as optical image). GCDs from Output 1 were then classified using Process 2, and CVs were separately classified by Process 3, thereby providing the resulting prediction (Output 2: classified GCDs, and Output 3: classified CVs) of either battery or pseudocapacitor with a percentage confidence rating of 0–100%, while the errors were monitored and minimized to improve the prediction. Here, the capacitive tendency (0–100%) was first defined by the percentage confidence value, indicating the probability of CV shape as peak (0% capacitive tendency) and box shape (100% capacitive tendency). In the final step (Fig. 3b), the classified CVs (in Output 3) were labeled according to four percentage confidence classes—100% battery, 50% battery, 50% pseudocapacitor and 100% pseudocapacitor—before being further modeled in Processes 4 and 5 to provide the capacitive tendency based on a percentage confidence of 0–100%.

An alternative way to understand the definition of capacitive tendency is to analyse it as the deviation from the ideal of the purely capacitive signal (is easy to recognize). When the trained model is confident that the curve is close to a rectangle (for CV) or a triangle (for GCD), then this implies that the curve is close to an ideal capacitive signal. On the contrary, a curve whose confidence value is close to zero means that the curve has a different contribution. Basically, the capacitive tendency reflects the analysis of the signal shape. It is

information based on a geometric shape. Of course, alternatives could be used. However, the use of the classical formalism, as indicated in the "ideal CVs" area in Fig. 1a, is impossible when the shape of the electrochemical signal deviates from this ideal. In the purely mathematical domain, the possibility of adding a rectangle to a closed geometric shape (a CV is a closed geometric shape) is a complex mathematical situation. Thus, our data science-driven by supervised deep learning approach is a suitable alternative.

## Results and discussion
This section explains how the models for CV and GCD classification were established for this specific dataset through the validation of different CNN architectures. The selection was based on well-known parameters including Accuracy, Sensitivity, Specificity, Accuracy, and F-Score. Moreover, the most accurate model was developed for use as the descriptor in order to determine the capacitive tendency of the various electrochemical behaviors, by applying the experimental data of various electrode materials. Ultimately, the selected model is destined for use by electrochemists as a tool for determining the nature of their materials.

### Validation of architectures
To select the Convolutional Neural Network architecture best suited to our datasets, the validation of a total of five models (ResNet50,

MobileNetV2, VGG16, Xception, and 8-Layer CNN) was first performed using Processes 2 and 3 with different types of input and output (Table 1). These architectures were chosen based on the reported accuracy ranking ascribed to the models' performance from ImageNet validation[35,36]. In this step, the prediction was governed by binary classification to obtain only two different outputs, namely (i) battery or

(ii) pseudocapacitor, since the model had been trained and supervised with CV and GCD datasets without ambiguity. ResNet50 was found to be the most accurate and precise one out of all the models (Table 1, Supplementary Figs. 7–11) and was thus selected to further prediction in the next step. Moreover, ResNet50 is more adapted to the variety of data that will be input by the users, for example, plot with different frame and font styles and different color curves.

To demonstrate the efficiency of the model, 5598 CVs and 2979 GCDs were randomly selected and entered the classifier according to Processes 2 and 3. Supplementary Fig. 12 clearly demonstrates that the majority of predicted datasets showed a 100% confidence rating, which would suggest that our ML model displays a high level of precision and reliability with a negligible risk of error.

### Table 1 | Performance of different architectures for classification

| CNN-model | Accuracy (%) | Sensitivity (%) | Specificity (%) | Precision (%) | F1-Score (%) |
|---|---|---|---|---|---|
| *GCD Classification* | | | | | |
| **ResNet50** | 94.22 | 93.84 | 94.45 | 94.16 | 93.99 |
| **MobileNetV2** | 93.11 | 92.56 | 93.07 | 93.12 | 92.82 |
| **VVG16** | 92.22 | 92.24 | 94.61 | 91.78 | 91.99 |
| **Xception** | 93.77 | 93.98 | 96.49 | 93.33 | 93.60 |
| **8-Layer CNN** | 94.00 | 93.73 | 94.77 | 93.82 | 93.78 |
| *CV Classification* | | | | | |
| **ResNet50** | 95.80 | 93.52 | 96.74 | 94.65 | 94.07 |
| **MobileNetV2** | 94.64 | 92.62 | 96.42 | 93.24 | 92.92 |
| **VVG16** | 94.36 | 93.12 | 97.13 | 91.53 | 92.28 |
| **Xception** | 93.04 | 88.87 | 94.35 | 91.31 | 90.00 |
| **8-Layer CNN** | 93.65 | 89.08 | 94.26 | 92.77 | 90.74 |

GCD and CV classification comparison based on evaluation values obtained from five different architectures: ResNet50, MobileNetV2, VGG16, Xception, and 8-Layer CNN.

### Validation of theoretical CVs and GCDs

In this part, the simulations of CV and GCD images were done using basic equations from theoretical electrochemistry including Faradaic process with peak-shaped CV[37], and EDLC with box-shaped CV which relies on Eqs. 2 and 3. The simulated images were then classified by the trained model (process 4–5). The equation for CVs showing redox peaks is given as follows:

$$\frac{i}{i_{\max}} = \frac{e^{\frac{F}{R \cdot T}\left(E - E_{peak}^0\right)}}{1 + \left(e^{\frac{F}{R \cdot T}\left(E - E_{peak}^0\right)}\right)^2} \tag{2}$$

where $\frac{i}{i_{\max}}$ is the normalized current of the peak current function, $F$ is the Faraday constant, $R$ is the gas constant, $T$ is the temperature, $E$ is

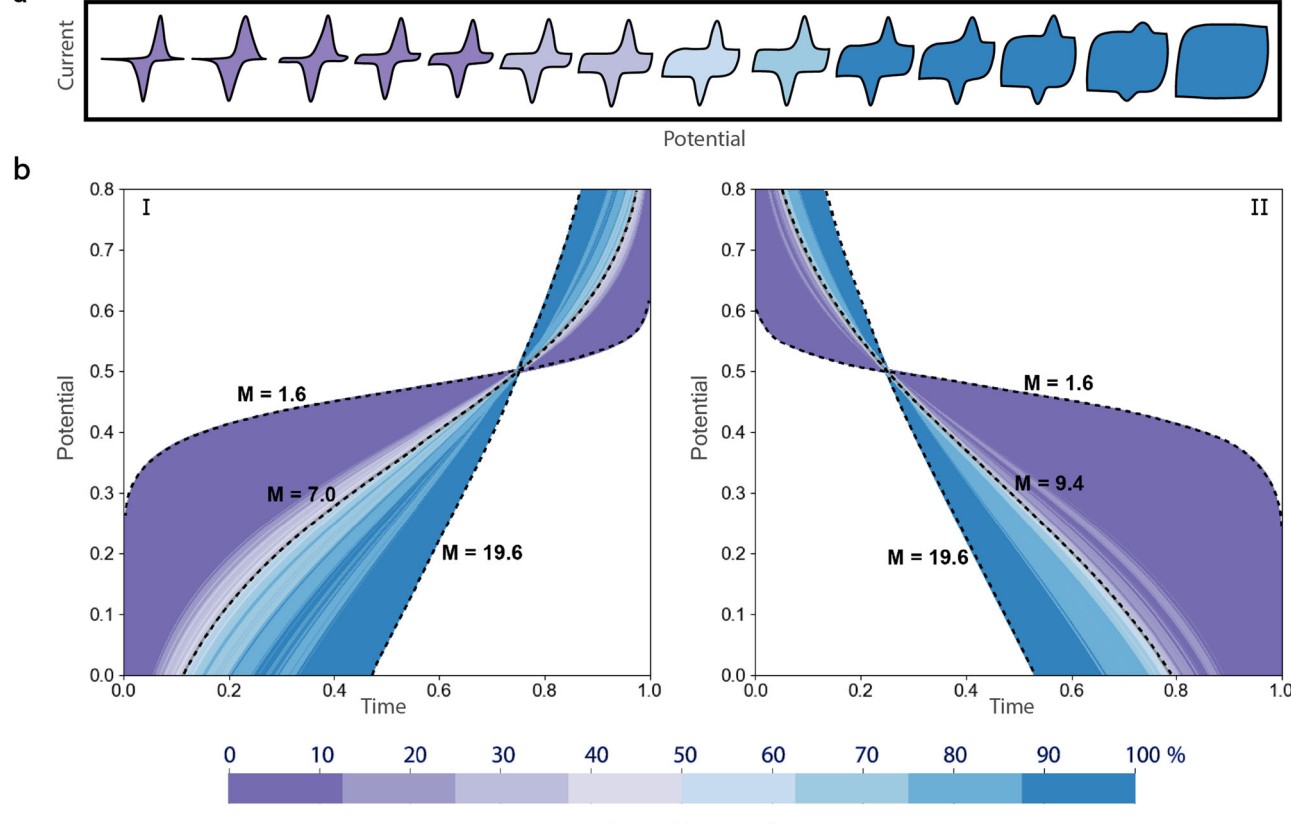

**Fig. 4 | Theoretical calculations and capacitive tendency of CV and GCD curves.** The illustration of (**a**) classified theoretical CVs with Gaussian and box shapes as the components, and (**b**) classified theoretical galvanostatic charge (I) and discharge (II) curves obtained by using Eq. 4. with a varying M parameter. The color of each curve is related to the probability of being battery (purple gradient bar) or capacitive material (blue gradient bar).

the applied potential and $E_{peak}^0$ is the peak potential. The box-shaped *EDLC* current function is given by:

$$\frac{i}{i_{max}} = 1 - e^{-\frac{t}{R \cdot C}} \qquad (3)$$

where $C$ is the capacitance, $R$ is the resistance and $t$ is the charging period[38]. It was shown that capacitive behavior is more pronounced the further the *CV* shape deviates from peaked to rectangular (Fig. 4a).

Furthermore, simulating number of theoretical *GCD* images with the transition in curvature from straight to plateau feature could be applied with the classification model (process 2) in order to see the region of ambiguity. Using Eq. 4 by varying M parameter:

$$E = M \cdot \left(\frac{R \cdot T}{n \cdot F}\right) \log\left(\frac{\sqrt{\tau} - \sqrt{t}}{\sqrt{t}}\right) + E_{\tau/4} \qquad (4)$$

where $E$ is the potential, $n$ is the number of electron transfers, t is the charging/discharging time, τ is the time constant, $E_{\tau/4}$ is the quarter-wave potential and $M$ is the mathematical factor permitting the manipulation of the galvanostatic curve to show either a plateau feature (as found in battery material measurements) or straight line (as in supercapacitor material measurements), the continuum GCD curves were obtained, as shown in Fig. 4b (blue, gray, and purple lines).

Figure 4b(I) shows that a battery-type signature was found to apply for an $M$ value range of between 1.6 and 7 (purple zone, with a 90-100% confidence rating), whereas the prediction point to a pseudocapacitor-type for $M$ values of between 7.1 and 19.6 (blue zone, with a 70-100% confidence rating). Similarly, this result was also observed for theoretical discharging profiles, as shown in Fig. 4b(II). However, in the gray zone when M is around 7.0 during charge and 9.4 during discharge, respectively, the predictor was hesitant to define the

signal type, suggesting that a certain ambiguity occurs when the curvature of the *GCD* signal is somewhere between a straight line and a plateau, as has already been observed and which is consistent with experimental measurements related to pseudocapacitive materials (Fig. 5c). The most pertinent conclusion that can be drawn from this calculation is that our model demonstrated the transition region of *GCD* signals in accordance with the continuum transition concept as proposed by Fleischmann et al.[16]. Our model clearly demonstrates the source of the confusion for both humans and computers, which stems from the fact that these behaviors all originate from faradaic processes where electron transfer is the elementary step. This explains why the results of theoretical studies only hold true for basic scenarios. More complex behaviors, however, are frequently observed in experimental measurements and account for vast amounts of data.

## Revealing the nature of electrode materials through supervised machine-learning

In accordance with the main purpose of this study, namely overcoming human limitations when it comes to understanding electrochemical signals, the objective in this section concerned clarifying the behavior of faradaic electrode materials. To this end, experimental CVs from Fig. 1 were applied to the model to predict the capacitive tendency behavior of various electrode materials that conventionally can be calculated from dQ/dV = constant in only simple cases such as supercapacitor materials but could be too complex to apply for pseudocapacitors. Well-known pseudocapacitive and battery materials from the literature, such as $MnO_2$ and NMC, were compared not only to separate the signals produced by Processes 2 and 3 according to the conventional binary classification, but also to establish a new standard that we called capacitive tendency. Processes 4 and 5 broadened the classification range to create a statistical tendency representing an interpretable value: in the range of 0% denoting a battery, to 100% being a

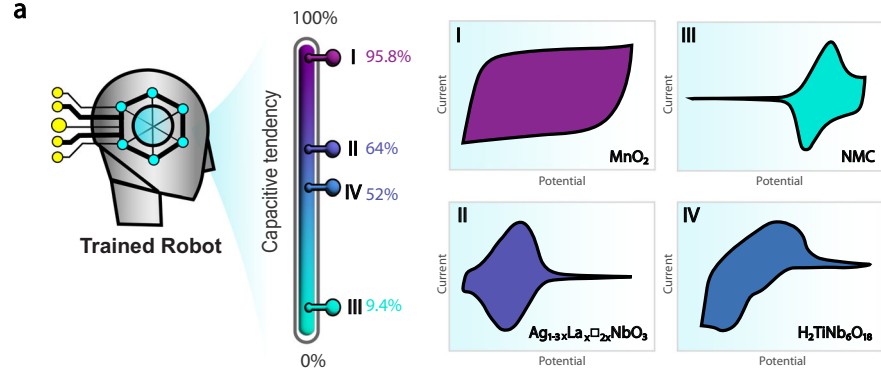

**Fig. 5 | Prediction of capacitive behavior of various electrode materials.** The capacitive tendency prediction of experimental voltammograms of (**a**) the well-known pseudocapacitor and battery electrode materials $MnO_2$[49], and NMC[50],

compared with the ambiguous CVs of $Ag_{1-3x}La_x\square_{2x}NbO_3$[44], and $H_2TiNbO_{18}$[43]. The predicted (**b**) CVs and (**c**) GCDs of other electrode materials from the literature, as mentioned in Fig. 1.

pseudocapacitor. Finally, we were able to predict the capacitive behavior of various electrode materials from experimental data, as demonstrated in Fig. 5.

As previously mentioned, the exemplary rectangular and peak shapes are unfortunately not often present when it comes to systems exhibiting fast charge/discharge behavior or when pseudocapacitive materials are investigated. Electrochemists thus find it difficult to analyze the voltammograms correctly in the face of such a variety of shapes, with even the CVs of $V_2C$, $Nb_2O_5$ and nano-$MoS_2$ electrode materials (Fig. 5b) displaying a similar capacitive tendency of around 52–53%. This finding served to emphasize the necessity of using machine-learning as a decisive tool for interpreting CV signals displaying a complexity that is beyond human discernment. The understanding of the origin of the electrochemical behavior is the key point for the deep knowledge and for the future development of the electrode materials. TB robots have been used to study the physicochemical features of a variety of electrode materials, including carbon electrodes, MOFs, COFs, graphite, NMC, and MXenes materials, determining the capacitive tendency of the CV in 'continuum region from the recent papers (as shown in Supplementary Fig. 21).

### The limitation of the binary classification battery vs. pseudocapacitor

During this phase of our research, numerous scientific articles containing the keyword "battery" (2011 articles) or "pseudocapacitor" (1346 articles) were analyzed using our supervised ML model to provide a statistical analysis of the number of papers containing a keyword that was in contradiction to their signals (used articles outside the training dataset). Briefly, the articles were randomly selected and their relevant CV and GCD signals were extracted and then simply classified into either battery or pseudocapacitive type using only Processes 2

and 3. The outputs in Fig. 6 depict that around 67% of the papers with a "pseudocapacitor" keyword are consistent with their experimental observations. Unexpectedly, however, nearly 50% of the articles with a "battery" keyword displayed contradicting signals. These results serve to reinforce the fact that human-based interpretation could greatly benefit from being replaced with computing techniques such as *ML*. Apparently, our machine-learning classification technique showed the significant portion of the articles using binary keywords (battery or pseudocapacitor) that contradict (mismatched) with their electrochemical signal (Supplementary Information).

This result shows perfectly the limit of the binary approach in the field. Because analysing a binary classification leads to this misclassification by the authors. Our approach, using capacitive tendency, allows a unification of the measurements, by including them in a "spectrum" as proposed by Fleischmann et al.[16].

### Online tool kit for CV/GCD classification

In order to facilitate the task of users worldwide when it comes to classifying the electrochemical behaviors (battery or pseudocapacitor) of their experimental data (CVs and GCDs), we have launched an online tool for analyzing these signals and providing an output in the form of a capacitive trend (or percentage confidence rating). It is publicly available at http://supercapacitor-battery-artificialintelligence.vistec. ac.th, and details are also provided in the Supplementary Information part 8---the website description.

The training database boundary is fixed using only scientific data with the pseudocapacitor or battery keywords associated. It is recommended for reader to use the present model to compare signal associated to EDLC, pseudocapacitor and Metal-ion battery. The present model isn't adapted to redox flow battery, and fuel-cells. Moreover, the present model doesn't provide any performance predictions. The typical useful application is to compare the same family of

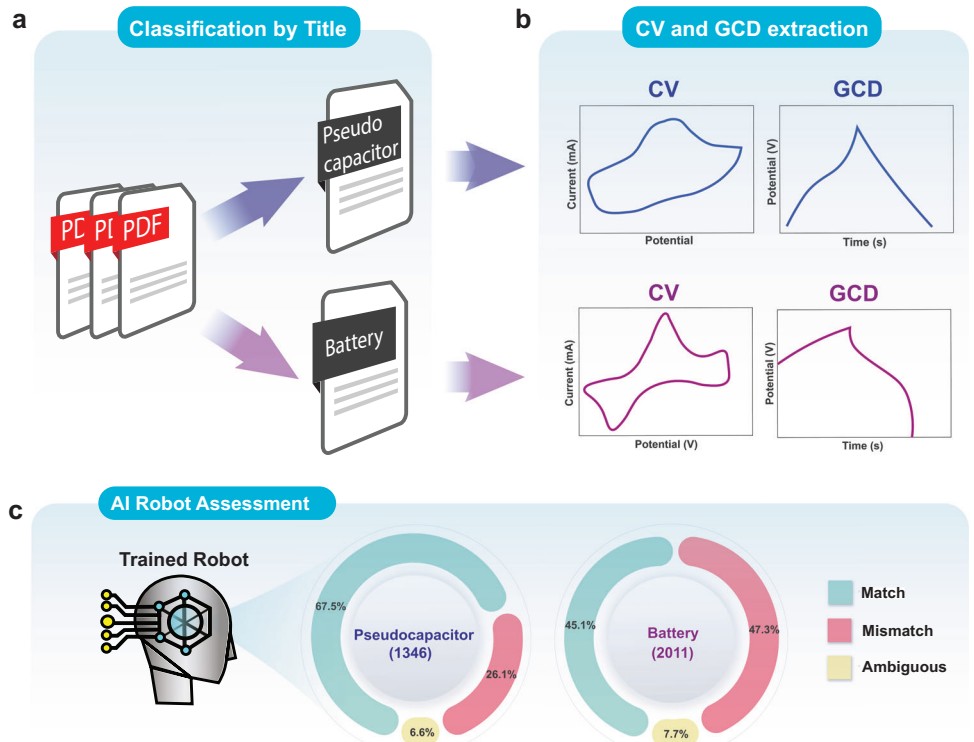

**Fig. 6 | Comparison of paper definitions and predicted results. a** The methodology behind the title classification of papers as either a battery or pseudocapacitor, followed by **b** CV and GCD extraction and then **c** the matched/mismatched outputs using our classifiers (Processes 1, 2 and 3). The percentage correlation between titles for pseudocapacitor and battery materials vs. correctly classified CVs and GCDs.

materials (i.e, MOF, NMC, MXene) but presenting a different electrochemical behavior. That is the generality and universality of this study.

The research presented herein has successfully managed to resolve the decades-old conundrum concerning the interpretation of electrochemical signals from *CVs* and *GCDs* by making full use of advanced computing technology in order to classify the behavior of materials as battery-like or pseudocapacitor-like. Specifically, we demonstrated that supervised ML is a powerful and accurate way to distinguish between these often complex signals. Our study also highlights the recurrent issue of the titles of scientific papers often contradicting the results of their own data, especially when it comes to those articles with "battery" in the title. This demonstrates the superiority of machine learning over human-based analysis for the interpretation of electrochemical signal images. Machine-learning is able to quickly and accurately transform the shape information of images into predicted values, while human-based analysis is far slower and more subjective. This is due to the fact that machine learning algorithms are able to learn from large datasets of images and extract patterns that are not visible to the human eye. As a result, machine learning is a more reliable and objective approach to the analysis of electrochemical signal images. As a major contribution to our peers in the electrochemical energy storage community, we are delighted to announce a first online tool based on our model toward simple CV and GCD image classification via our precise marker, called capacitance tendency (quantitative information presented in percentage), affording them the possibility of a quick and easy standard to refer to when attempting to determine the nature of their new materials. Using the present program, all experimental user will be able to correlate chemical information to capacitive tendencies, as the scan rate, the current density. However, a potential drawback of the current classifier is that it can only predict the resistive tendency of electrochemical signals based on CV/GCD image data. A more comprehensive classifier by featuring text-mining of material information of a hidden information such as labels, scan rate, electrolytes in the figure could be an ultimate strategy for future perspectives on artificial intelligence for energy storage technology.

## Data availability
The figures, tables and literatures data of capacitive performance generated in this study are provided in the Supplementary Information.

## Code availability
Machine-learning models and datasets are made publicly available at GitHub repository[27]. The instruction is provided in both supporting information and on Github repository.

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

## Acknowledgements

O.F thanks Institut Universitaire de France for the financial support. Y.Z, J.D., and O.F. thank Chengdu University for the collaboration support. Website hosting is supported by VISTEC server. This work is supported by funding from Thailand Science Research and Innovation (TSRI) (Grant No. FRB660004/0457). VISTEC thanks for the impressive technical support of Assistant Researcher Konthee Boonmeeprakob in the Python program. O.F. gives a special thanks to Fred Favier, Institut Charles Gheradht Montpellier, for the long discussion about the concept of pseudocapacitor.

## Author contributions

J.D., Y.Z. had centralized the dataset of scientific papers, checked the output, and tested the trained model. E.C., O.C., T.B. tested the trained model, checked the dataset, and sent the experimental data. S.D., O.F. write the papers. S.D. programed Python and trained the model.

## Competing interests

The authors declare no competing interests.
