## [Peer Review File · Nature Communications]

Capacitive tendency concept alongside supervised machine-learning toward classifying electrochemical behavior of battery and pseudocapacitor materialsEditorial Note: Parts of this peer review file have been redacted as indicated to maintain the confidentiality unpublished data.

Reviewer #1 (Remarks to the Author):

Reviewer's general comment: The work focused on a machine learning method with the capacitive tendency for classifying battery and pseudocapacitor materials. The manuscript is within the scope of the Journal. To help improve the paper's quality, my suggestions and comments are shown below.

1) Abstract:

(1) It is suggested to give a brief description of the current research progress in the forms of the classification of battery and pseudocapacitor electrode materials, especially the efforts in machine learning. Then, please summarize the research gaps, as well as the motivations and advantages of your method.

(2) It is suggested to introduce the mechanism of the proposed methodology in one to two sentences.

(3) Qualitative results with quantitative data are necessary to support the contribution of the work.

2) Introduction:

(1) The definition of capacitive tendency is not clear in the "introduction part", while the advantages are given. The mechanism of "capacitive tendency" to help classify battery and pseudocapacitor materials is suggested to be further explained.

(2) Introduction: as mentioned in the manuscript, the text-mining algorithms have been developed to efficiently extract various specific information of the materials, like BatteryDataExtractor and Li-ion battery annotated corpus. In this study, the authors applied ML for electrochemical signal interpretation. This belongs to the first application. From this point, the innovation or originality might be questionable.

(3) Original contribution at the end of Introduction needs to be further enhanced.

(4) Compare your approaches used in your study to the others in terms of their advantages and drawbacks.

(5) The proposed algorithm should be tested on different data sources.

3) Methods:

(1) Dataset Construction: according to the paper, 80% of data is used for training, while 20% of data is used for validation. Here the "validation" may be changed into "test". The "validation" in machine learning is part of the training set, which is used for hyperparameter optimization and model architecture optimization, while the test set is used for model performance evaluation based on data out of the training set.

(2) Machine-learning for CV/GCD classification procedures: according to the “alternative way to understand the definition of capacitive tendency”, the capacitive tendency is an index that can quantify the difference between the theoretical curves and the actual curves and help tell the classification of the targeted material. However, many methods can be used to quantify the difference between the two curves. Only qualitative description is not convincing. Hence, please explain the necessity of CNN models and give the reference or quantitative comparison results to convey the superiority of the CNN model over traditional methods.

(3) In addition to the detailed mathematical descriptions of methods adopted in the proposed algorithm, the motivations, and reasons why you choose the methods are suggested to be added in detail.

4) Results and Discussion:

(1) In Figure 4, the formula and citation format should be corrected. The same mistakes can be found in Figure 6.

4) Conclusion

(1) point-by-point items with quantitative results will be more effective to convey the main findings of this study.

(2) As mentioned in Conclusion, ‘ML application in distinguishing between these often complex signals’, how to distinguish the database for training, testing and validation?

Other questions:

(1) Supervised ML is a powerful tool for fast and accurate calculation, while the main issue is its poor capability in new knowledge exploration. In other words, it is good at knowledge exploitation within the training database, but fails to classify materials and predict the performance out of the training database boundary. How to address these issues? More discussion on the drawbacks will be wonderful.

Overall, the topic of this study is important. Hope the comments can be helpful to improve the paper’s quality.

Reviewer #2 (Remarks to the Author):

This Manuscript provides an incredibly useful tool for estimating the percentage of capacitive or battery-like behavior of energy storage materials. Due to the very large number of publications on pseudocapacitive materials and confusion about the appropriate classification, the free-accessible robot tool provided by the group of Olivier Fontaine can be of large interest to the research community.

Nevertheless, often the shape of the cyclic voltammetry or the galvanostatic profile may become similar to "pseudocapacitive" at high scan rate, high current, respectively. How the robot takes in consideration the capacitive tendency with respect to the scan rate or current used?

Reviewer #3 (Remarks to the Author):

The article titled "A Novel Approach for Classifying Battery and Pseudocapacitor Materials Using Capacitive Tendency and Supervised Machine Learning" discusses the use of supervised machine learning techniques to analyse, interpret and classify electrochemical signals in energy storage devices (batteries and supercapacitors). The use of supervised machine learning and the development of an online tool for classification are significant contributions to the field, despite they appear more suitable for a methodology or computational journal rather than for an interdisciplinary one. While the article presents interesting findings and potential applications, it has both major and minor issues that should be addressed.

Major issues:

- 1) The article recalls the concept of a "continuum spectrum" to describe the transition between capacitive and battery-type signals. However, the authors acknowledge that this concept lacks mathematical support and is merely a postulate. This weakens the scientific rigor of the proposed approach and calls into question the reliability of the findings. The Authors are encouraged to argue on this possible weakness.
- 2) The validation of the classification architectures used in the machine learning approach is not adequately explained. The article mentions benchmarking the models based on five metrics but does not provide sufficient details or results to assess the performance of the selected models in the main manuscript (only in the Supplementary Information). Please include a picture with main prediction performance for all Processes during both training and validation steps.
- 3) The article briefly mentions the use of computing techniques and text mining in energy storage research but fails to provide a comprehensive comparison with existing techniques for interpreting

electrochemical signals. This limits the understanding of how the proposed machine learning approach contributes to the field and whether it outperforms or complements existing methods. Have you conducted any comparative analysis with existing methods to demonstrate the superiority of the capacitive tendency metric? It would be valuable to provide a quantitative comparison and discuss the advantages of your approach over conventional classification techniques in terms of both computational requirements and accuracy.

Minor issues:

1) The article lacks sufficient contextualization within the broader field of energy storage research. While the article discusses the significance of distinguishing between capacitive and battery-type signals, it does not sufficiently connect this research to the current state of the field or highlight how the findings contribute to advancing energy storage technologies. For instance, the use of machine learning approach to analyse the impact of physicochemical features of carbon electrodes on the capacitive performance of supercapacitors could be mentioned.

2) In the article, you mentioned that there can be an overlap between battery and pseudocapacitor signals due to their faradaic nature. It would be beneficial to discuss some examples or case studies where this overlap occurs and how the proposed capacitive tendency metric effectively distinguishes between them. This would provide concrete evidence of the robustness and accuracy of your approach.

3) Could you please elaborate on the dataset used for training the supervised machine learning algorithm? Specifically, how diverse is the dataset in terms of electrode materials and electrochemical behaviours? It would be helpful to understand the representativeness of the dataset and its impact on the performance and generalizability of the proposed classification approach.

4) The article lacks clarity in explaining the methodology used for dataset construction and the specific processes involved in the machine learning classification. Important details such as the selection criteria for training and validation datasets, data preprocessing techniques, and hyperparameter tuning are not sufficiently explained. This hinders reproducibility and makes it challenging for readers to evaluate the methodology. Moreover, in the Methods section, all Processes and Outputs shall be described in chronological order.

5) How GCD and CV pictures are extracted from articles and recognized among other Figures? Did you check that the adopted measure units are the same in these graphs? How did you train Process 1? How did you label the GCD and CV pictures so that they belong to one of these two classes? How did you label the Output 3 into the three types of training sets for Process 3 and 4?

6) “Moreover, cross-validation was performed with the experts in the field with the number of meetings”: what do you mean? Which types of cross-validations were performed? Who were the experts? Please describe such cross-validation with more quantitative arguments.

7) Please better describe why two models (Process 4 and 5) have been trained to predict the capacitive tendency if the output figure of merit is just one. In this sense, Output 4 and 5 seem redundant (they should be complementary with each other).

8) The subsection “The issues surrounding electrochemical signal identification” appears as a repetition of Introduction rather than a Results. Please improve the readability of the article by removing redundant parts.

9) In the analysis carried out in Figure 7, were the considered articles outside training set?

10) The article briefly mentions the limitation of electrochemical signals deviating from ideal curves, but it does not extensively discuss other potential limitations of the proposed machine learning approach. Furthermore, the article does not provide a detailed discussion on future directions for improving the methodology or addressing these limitations.

11) Please improve English language and correct typos (e.g., caption of Fig. 6, table headings of Fig. 6).

Responses to Comments raised by Reviewers

**“A Novel Approach for Classifying Battery and Pseudocapacitor Materials
Using Capacitive Tendency and Supervised Machine Learning”**

No.: NCOMMS-23-20980

For Editor:

Dear Editor and Reviewers,

You will find, in black numerated, the reviewer comment and in blue with this style our explanation

-----Inserted in manuscript at P.XX Line XX-----

In blue and italic like here

-----The content above resides within the manuscript-----

Reviewer 1

Reviewer's general comment: The work focused on a machine learning method with the capacitive tendency for classifying battery and pseudocapacitor materials. The manuscript is within the scope of the Journal. To help improve the paper's quality, my suggestions and comments are shown below.

Thank you for your detailed and helpful comments on our manuscript. We have carefully considered your comments and suggestions, the reviewer 1 suggestions improve a lot the present manuscript.

Abstract:

1. It is suggested to give a brief description of the current research progress in the forms of the classification of battery and pseudocapacitor electrode materials, especially the efforts in machine learning. Then, please summarize the research gaps, as well as the motivations and advantages of your method.

Thank you for your comments, and we emphasized more on this point in the Abstract, here the change:

-----Inserted in manuscript at P.2 Line 22-32-----

“Up to date, the difficulty lies in determining which group these materials fall into one or another electrode type (battery vs. pseudocapacitor) through simple binary classification as black and white. Recently in the field of energy storage, the famous qualitative term ‘continuum transition’ has been introduced ascribing to the overlapping of the characteristic of electrochemical signals between battery and pseudocapacitor. To overcome this conundrum, we applied supervised machine-learning of image classification towards the electrochemical shape analysis (over 5,500 CVs and 2,900 GCDs) and the confidence percentage of the prediction reflects the shape tendency of the curves, consequently defined as the new maker called “capacitive tendency”. This predictor not only surpasses the limitations of human-based classification but also provides statistical tendencies regarding electrochemical behavior.”

-----The content above resides within the manuscript-----

2. It is suggested to introduce the mechanism of the proposed methodology in one to two sentences.

We added the mechanism of the proposed methodology in the Abstract.

-----Inserted in manuscript at P.2 Line 27-30-----

“To overcome this conundrum, we applied supervised machine-learning of image classification towards the electrochemical shape analysis (over 5,500 CVs and 2,900 GCDs) and the confidence percentage of the prediction reflects the shape tendency of the curves, consequently defined as the new maker called “capacitive tendency”.”

-----The content above resides within the manuscript-----

3. Qualitative results with quantitative data are necessary to support the contribution of the work.

Thank you for your comments, we put the number of data used in this work.

-----Inserted in manuscript at P.2 Line 27-29-----

“To overcome this conundrum, we applied supervised machine-learning of image classification towards the electrochemical shape analysis (over 5,500 CVs and 2,900 GCDs)”.

-----The content above resides within the manuscript-----

Introduction:

4. The definition of capacitive tendency is not clear in the “introduction part”, while the advantages are given. The mechanism of “capacitive tendency” to help classify battery and pseudocapacitor materials is suggested to be further explained.

The “capacitive tendency” is defined based on the percentage confidence of the classification between box shaped and peak shaped CV. The percentage confidence ranges from 0-100% that

the CV is either classified as battery or pseudocapacitor. For example, the CV is classified as pseudocapacitor with 30 % confidence, it simply means that the CV has 30% capacitive tendency. Here the change inside the manuscript:

-----Inserted in manuscript at P.5 Line 99-105-----

“So, by this approach, we proposed the new definition call “capacitive tendency” based on the percentage confidence of the classification between box shaped and peak shaped CV, implying the capacitive behavior of electrode materials.”

-----The content above resides within the manuscript-----

5. Introduction: as mentioned in the manuscript, the text-mining algorithms have been developed to efficiently extract various specific information of the materials, like BatteryDataExtractor and Li-ion battery annotated corpus. In this study, the authors applied ML for electrochemical signal interpretation. This belongs to the first application. From this point, the innovation or originality might be questionable.

Previous approaches to data mining of electrochemical, chemical, and physical properties of energy storage materials have focused on text mining or material extraction. Despite the importance of the work, these approaches do not analyze electrochemical signal shape or interpretation.

Our work is the first to use supervised machine learning to interpret electrochemical signal shape, specifically CV and GCD images. This is a significant advancement because the shape of CV and GCD images can tell us about the underlying mechanism of the energy storage material, such as battery, pseudocapacitor, or supercapacitor. To clarify the originality, we rearranged the text in this part in the introduction:

-----Inserted in manuscript at P.4 Line 78-101-----

Recently, text-mining algorithms have been developed to efficiently extract various specific information of the materials from the article such as BatteryDataExtractor using bidirectional-encoder representations from transformers (BERT),^[1] and Li-ion battery annotated corpus

(LIBAC) based on ML, natural language processing (NLP), Named Entity Recognition (NER).^[2-4] However, the direct interpretation of the image data from figures remain difficult using the above method of data-mining from image. Machine learning has been used to predict the electrochemical mechanism involved in the reaction that expresses through a cyclic voltammogram (CV). Deep learning has also been used to distinguish the mechanism of the electrochemical reaction from CV based on residual neural network (ResNet) architecture. However, the application of machine learning to analyze electrochemical signals in the field of energy storage is still in its early stages and focused on analytical or fundamental electrochemistry.

This study presents the first time that electrochemical signal analysis (CV and GCD) has been performed using a machine learning (ML) approach based on image classification. This approach is well-suited for unlabeled data, noise-tolerant, and capable of handling complex data. Ultimately, this led to the determination of the capacitive behavior of electrode materials from thousands of scientific papers. The originality of this work lies in its use of machine learning (ML) to quickly and accurately interpret electrochemical signal images and transform them into accurate values. This is made possible by the large database of electrochemical energy storage images that is available to the ML model. This approach overcomes the limitations of human ability to interpret data, which can be too complicated in most cases. (Figure 1). So, by this approach, we propose the new definition call “capacitive tendency” based on the percentage confidence of the classification between box shaped and peak shaped CV, implying the capacitive behavior of electrode materials.”

-----The content above resides within the manuscript-----

6. Original contribution at the end of Introduction needs to be further enhanced.

We rewritten the text on the originality of the work at the end of Introduction.

-----Inserted in manuscript at P.5 Line 90-101-----

“This study presents the first time that electrochemical signal analysis (CV and GCD) has been performed using a machine learning (ML) approach based on image classification. This approach is well-suited for unlabeled data, noise-tolerant, and capable of handling complex

data. Ultimately, this led to the determination of the capacitive behavior of electrode materials from thousands of scientific papers. The originality of this work lies in its use of machine learning (ML) to quickly and accurately interpret electrochemical signal images and transform them into accurate values. This is made possible by the large database of electrochemical energy storage images that is available to the ML model. This approach overcomes the limitations of human ability to interpret data, which can be too complicated in most cases. (Figure 1). So, by this approach, we propose the new definition call “capacitive tendency” based on the percentage confidence of the classification between box shaped and peak shaped CV, implying the capacitive behavior of electrode materials.”

-----The content above resides within the manuscript-----

7. Compare your approaches used in your study to the others in terms of their advantages and drawbacks.

The advantage of our approach is that the fast and simple analysis by only two steps (1) inserting CV or GCD image and (2) click ‘predict’ to obtain the predicted capacitive percentage without pre-processing required. The prediction can tell the tendency of the material to behave like battery or supercapacitor/pseudocapacitor types. However, the drawback can be the lack of data mining of the image data such as the plot label, scan rate, or electrolyte. This requires a huge work and human power to integrate image recognition and text mining out of the image data.

The advantage of other studies is that they can extract electrochemical, chemical, physical properties of the materials directly from the context but not to interpret the electrochemical signal. However, the drawback of this approach is the signal interpretation from the image data of plots such as electrochemical CV/GCD, directly.

We added I the manuscript of the sentences emphasizing this point:

-----Inserted in manuscript at P.5 Line 82-94-----

“However, the direct interpretation of the image data from figures remain difficult using the above method of data-mining from image. Machine learning has been used to predict the electrochemical mechanism involved in the reaction that expresses through a cyclic

voltammogram (CV). Deep learning has also been used to distinguish the mechanism of the electrochemical reaction from CV based on residual neural network (ResNet) architecture. However, the application of machine learning to analyze electrochemical signals in the field of energy storage is still in its early stages and focused on analytical or fundamental electrochemistry.

This study presents the first time that electrochemical signal analysis (CV and GCD) has been performed using a machine learning (ML) approach based on image classification. This approach is well-suited for unlabeled data, noise-tolerant, and capable of handling complex data. Ultimately, this led to the determination of the capacitive behavior of electrode materials from thousands of scientific papers.”

-----The content above resides within the manuscript-----

8. The proposed algorithm should be tested on different data sources.

We used three different types of data sources:

- (1) Dataset from literature:

The major source of image data of CV and GCD (over 5,500 CVs and 2,900 GCDs) were extracted from a number of scientific papers. The image datasets are on Github repository (https://github.com/ice555mee/TB-robot_code-data). Available for the reader.

- (2) Theoretical GCD and CV:

In the manuscript (in the part **Validation of theoretical CVs and GCDs**), we generated the images of CV and GCD using electrochemical equations (Eq. 2-4). The images of different CV and GCD were obtained by varying the shape parameter and finally producing the variation of CV and GCD shaped image data as shown in Figure 5 in the manuscript.

- (3) Experimental dataset from experimental data:

The last data source is from the experimental image of CV and GCD from co-authors (Prof. Thierry Brousse and his team). We added **Figure ESI 22**, and **ESI 23** in supporting information.

We added in the manuscript:

-----Inserted in manuscript at P.7 Line 131-135-----

“We used three data sources for their study of CV and GCD images. The first source was a large dataset of over 5,500 CVs and 2,900 GCDs extracted from scientific papers. The second source was theoretical CVs and GCDs generated using electrochemical equations. The third source was experimental CVs and GCDs from co-authors.”

-----The content above resides within the manuscript-----

Methods:

9. Dataset Construction: according to the paper, 80% of data is used for training, while 20% of data is used for validation. Here the “validation” may be changed into “test”. The “validation” in machine learning is part of the training set, which is used for hyperparameter optimization and model architecture optimization, while the test set is used for model performance evaluation based on data out of the training set.

We changed the word from “validation” to “test”, since the validation was performed by human expertise validation to the ground truth (edited in manuscript at P.7 Line 125). So, the testing data was different dataset from the training dataset and only used to evaluate the performance of the testing model.

10. Machine-learning for CV/GCD classification procedures: according to the “alternative way to understand the definition of capacitive tendency”, the capacitive tendency is an index that can quantify the difference between the theoretical curves and the actual curves and help tell the classification of the targeted material. However, many methods can be used to quantify the difference between the two curves. Only qualitative description is not convincing. Hence, please explain the necessity of CNN models and give the reference or quantitative comparison results to convey the superiority of the CNN model over traditional methods.

Our defined ‘capacitive tendency’ is not for qualifying the difference between the theoretical curves and the actual curves but to tell the tendency of any (either theoretical or experimental)

CV or GCD curves to be more of the battery or supercapacitor characteristic shapes (box vs peak shape for CV).

And the capacitive tendency is not only the qualitative description but quantitative one based on statistic of the ML classification result that the percentage of confidence of the prediction of the CV shape reflects the probability of the shape of curves, basically like a tendency.

This research gap is about the shape interpretation of electrochemical curves presented in image data form. In conventional way, curve fitting using electrochemical equations is commonly done. But the problem is when the data is too big, for example, over 5000 CVs that need forever to fit all curves by human. Not only, when all these curves are in image form (JPEG, PNG, etc..), the value of all data points need to be extracted from the curve in the image at the first step and only this step need a big human power (number of people and time consuming).

So, only ML for image classification (such as CNN model) is the best way out for the analysis of a large number of images, faster and more accurate than any conventional methods. Nowadays, artificial intelligence is the most powerful approach in terms of performance, time consuming, cost, and resources for calculation, classification big data analysis.

Here, the model was applied to analyze MOF materials with different types of ligands (future work) as shown in Figure R1.

Figure R.1. TB robot application on classification of MOF materials with different structures.

We have calculated preliminary results using a different method (future work) that employs the concept of capacitive tendency. However, without our descriptor, it is impossible to correlate these results with the chemical nature of the materials.

Figure R.2: List of materials classified according to %pseudo (calculated by numerical method). The Y-axis represents the number of redox centers per nm^3 and the X-axis represents the total number of electrons in the crystal structure.

We added in the manuscript:

----- Inserted in manuscript at P.5 Line 83-101-----

“Machine learning has been used to predict the electrochemical mechanism involved in the reaction that expresses through a cyclic voltammogram (CV). Deep learning has also been used to distinguish the mechanism of the electrochemical reaction from CV based on residual neural network (ResNet) architecture, and focused on analytical or fundamental electrochemistry. However, the application of machine learning to analyze electrochemical signals in the field of energy storage is still in its early stages.

This study presents the first time that electrochemical signal analysis (CV and GCD) has been performed using a machine learning (ML) approach based on image classification. This approach is well-suited for unlabeled data, noise-tolerant, and capable of handling complex data. Ultimately, this led to the determination of the capacitive behavior of electrode materials from thousands of scientific papers. The originality of this work lies in its use of machine

learning (ML) to quickly and accurately interpret electrochemical signal images and transform them into accurate values. This is made possible by the large database of electrochemical energy storage images that is available to the ML model. This approach overcomes the limitations of human ability to interpret data, which can be too complicated in most cases. (Figure 1). So, by this approach, we propose the new definition call “capacitive tendency” based on the percentage confidence of the classification between box shaped and peak shaped CV, implying the capacitive behavior of electrode materials.”

-----The content above resides within the manuscript-----

11. In addition to the detailed mathematical descriptions of methods adopted in the proposed algorithm, the motivations, and reasons why you choose the methods are suggested to be added in detail.

----- Inserted in manuscript at P.9 Line 165-179-----

“ResNet50 was exploited in different steps denoted as Processes 1, 2, 3, 4, and 5 (as summarized in Figure 3c) according to the types of inputs and outputs. All the images extracted from scientific papers were then categorized by Process 1 (ResNet50 model) which yielded Output 1, comprising GCDs, CVs and other images (such as optical image). GCDs from Output 1 were then classified using Process 2, and CVs were separately classified by Process 3, thereby providing the resulting prediction (Output 2: classified GCDs, and Output 3: classified CVs) of either battery or pseudocapacitor with a percentage confidence rating of 0 – 100 %, while the errors were monitored and minimized to improve the prediction. Here, the capacitive tendency (0 – 100 %) was first defined by the percentage confidence value, indicating the probability of CV shape as peak (0% capacitive tendency) and box shape (100% capacitive tendency). In the final step (Figure 3b), the classified CVs (in Output 3) were labeled according to four percentage confidence classes — 100 % battery, 50 % battery, 50 % pseudocapacitor and 100 % pseudocapacitor — before being further modeled in Processes 4 and 5 to provide the capacitive tendency based on a percentage confidence of 0 – 100 %.”

-----The content above resides within the manuscript-----

Results and Discussion:

12. In Figure 4, the formula and citation format should be corrected. The same mistakes can be found in Figure 6.

Thank you for your helpful correction. We have made the necessary changes to the formula and citation formats in Figures 4 and 6.

Conclusion:

13. point-by-point items with quantitative results will be more effective to convey the main findings of this study.

We emphasize more of the definition of the ‘capacitive tendency’ that is quantitative description based on statistics and mathematics on classification using ML approach. Our descriptor is the first tool in the energy storage community that can transform information in the CV and GCD images (qualitative information of shape identification) to the certain number as ‘capacitive tendency’ (quantitative information of predicted value by AI).

-----Inserted in manuscript at P.19 Line 347-353-----

“This demonstrates the superiority of machine learning over human-based analysis for the interpretation of electrochemical signal images. Machine-learning is able to quickly and accurately transform the shape information of images into predicted values, while human-based analysis is far slower and more subjective. This is due to the fact that machine learning algorithms are able to learn from large datasets of images and extract patterns that are not visible to the human eye. As a result, machine learning is a more reliable and objective approach to the analysis of electrochemical signal images.”

-----The content above resides within the manuscript-----

And,

-----Inserted in manuscript at P.19 Line 354-356-----

“a first online tool based on our model toward simple CV and GCD image classification via our precise marker, called capacitance tendency (quantitative information presented in percentage),”

-----The content above resides within the manuscript-----

14. As mentioned in Conclusion, ‘ML application in distinguishing between these often complex signals’, how to distinguish the database for training, testing and validation?

The database for training is the images of clear box or peak shaped CVs and the straight triangular or plateau shape GCDs, without ambiguity as representatively. For testing dataset, we used both ambiguous and non-ambiguous image data to test the performance of the model, where the performance was evaluated by Accuracy, Sensitivity, Specificity, Precision, and F1-score (**Table ESI 2**).

We added in the manuscript:

-----Inserted in manuscript at P.7 Line 121-128-----

*“In the training process of GCD (process 2) and CV (process 3) classification, CV and GCD images were firstly labeled as belonging to one of two classes, namely battery or pseudocapacitor following the criteria of non-ambiguous signal shape (which can be put into four categories: (1) Box shaped CV, (2) Peak shaped CV, (3) Triangular GCD, and (4) Plateau GCD) for 80 % of total data, where 20 % of total data was used as testing data. These training processes is based on binary classification of electrochemical signal, such as the box vs peak shaped CV, and the triangular vs plateau shaped GCD, as represented in **Figure ESI 6**, where all image datasets used are available on Github.”*

-----The content above resides within the manuscript-----

Other questions:

15. Supervised ML is a powerful tool for fast and accurate calculation, while the main issue is its poor capability in new knowledge exploration. In other words, it is good at knowledge exploitation within the training database, but fails to classify materials and predict the performance out of the training database boundary. How to address these issues? More discussion on the drawbacks will be wonderful.

While the training dataset that only contains those non-ambiguous images of CV or GCD, the testing dataset contains both ambiguous and non-ambiguous data to be used to test the classification, where our based model (ResNet50) gave more than 94.07 of F-score and more than 95 % accuracy (Table ESI 2). In our work, we only focus on training the model to distinguish between box and peak shapes for CV curve. In this case, we can use this prediction for understanding the origin of the electrochemical phenomena that can partially originate from redox reaction or electrochemical double layer capacitance. The result will give the possibility of the electrochemical behavior of the electrode materials. However, the other drawback that could be more useful for the better improvement or for the new classifier that to further predict in terms of other electrochemical information the resistive tendency of electrochemical signals.

We added in the manuscript:

-----Inserted in manuscript at P.19 Line 360-364-----

“However, a potential drawback of the current classifier is that it can only predict the resistive tendency of electrochemical signals based on CV/GCD image data. A more comprehensive classifier by featuring text-mining of material information of a hidden information such as labels, scan rate, electrolytes in the figure could be an ultimate strategy for future perspectives on artificial intelligence for energy storage technology.”

-----The content above resides within the manuscript-----

Overall, the topic of this study is important. Hope the comments can be helpful to improve the paper’s quality.

Thank reviewer 1 for their feedback on the paper. The comments are helpful in improving the quality of the paper.

Reviewer 2

This Manuscript provides an incredibly useful tool for estimating the percentage of capacitive or battery-like behavior of energy storage materials. Due to the very large number of publications on pseudocapacitive materials and confusion about the appropriate classification, the free-accessible robot tool provided by the group of Olivier Fontaine can be of large interest to the research community.

1. Nevertheless, often the shape of the cyclic voltammetry or the galvanostatic profile may become similar to “pseudocapacitive” at high scan rate, high current, respectively. How the robot takes in consideration the capacitive tendency with respect to the scan rate or current used?

Currently, the robot is not able to take into consideration the scan rate, high current, and other "chemical information" concerning the analysis of scientific papers extracted. This is because this part involves the use of text mining. However, future users of the robot can simply input the image of their CV produced at any scan rate or GCD at any current density. The predicted result (capacitive tendency) will relate to the shape of CV or GCD obtained from the experimental condition performed.

We added in the manuscript:

-----Inserted in manuscript at P.19 Line 358-360-----

“Using the present program, all experimental user will be able to correlate chemical information to capacitive tendencies, as the scan rate, the current density.”

-----The content above resides within the manuscript-----

Reviewer 3

The article titled "A Novel Approach for Classifying Battery and Pseudocapacitor Materials Using Capacitive Tendency and Supervised Machine Learning" discusses the use of supervised machine learning techniques to analyse, interpret and classify electrochemical signals in energy storage devices (batteries and supercapacitors). The use of supervised machine learning and the development of an online tool for classification are significant contributions to the field, despite they appear more suitable for a methodology or computational journal rather than for an interdisciplinary one.

Firstly, we would like to address the justification of the journal:

This work is significant for a large reader, as electrochemists, materials chemists, computing sciences, data sciences, because it has the potential to change the way that electrode materials are identified. By providing a more accurate and reliable method for understanding electrochemical behavior, our work will accelerate the development of new and improved energy storage technologies. We believe that this work is an interest for a large type of expertise.

While the article presents interesting findings and potential applications, it has both major and minor issues that should be addressed.

Major issues:

1. The article recalls the concept of a "continuum spectrum" to describe the transition between capacitive and battery-type signals. However, the authors acknowledge that this concept lacks mathematical support and is merely a postulate. This weakens the scientific rigor of the proposed approach and calls into question the reliability of the findings. The Authors are encouraged to argue on this possible weakness.

Our manuscript provides a quantitative approach to this postulate, as shown in Figure R.3, by introducing a metric or indicator to assess the degree of deviation from the ideal shape of a capacitor in the CV or GCD plots.

[redacted]

We understand that Reviewer 3 may have interpreted the **Figure R.3b(left)** (Figure 5 of ref [16]) as a model, but it is crucial to clarify that it is just a direction and a point of view, as the objective of a perspective paper, in mathematic notified as a conjecture.

To clearly identify our originality vs. the paper published in REF [16], we added in the manuscript, **Page 4**. And this quantification of the above mentioned figure from Ref[16] was demonstrated in **Figure ESI 21**, in the manuscript.

-----Inserted in manuscript at P.4 Line 66-70-----

“In order to complete this concept of ‘continuum spectrum’ and to provide the real quantitative value to it, we analyze the electrochemical signals with the help of supervised machine-learning for achieving the descriptor, “capacitive tendency” that allows our community to quantify this important spectrum.”

-----The content above resides within the manuscript-----

2. The validation of the classification architectures used in the machine learning approach is not adequately explained. The article mentions benchmarking the models based on five metrics but does not provide sufficient details or results to assess the performance of the selected models in the main manuscript (only in the Supplementary Information). Please include a picture with main prediction performance for all Processes during both training and validation steps.

The classification architectures were validated by evaluating their performance using the following metrics: accuracy, sensitivity, specificity, precision, and F1-score (see Table ESI 2). These metrics are commonly used in supervised machine learning, so we followed this convention. We further explained the validation of different architecture in the main manuscript and as well referred in the Supplementary Information such as loss curve in **Figure ESI 7- ESI 11**.

Following reviewer 3 recommendation, we also rearranged this part of information by putting Table ESI 2 in the main manuscript in Figure 3.

3. The article briefly mentions the use of computing techniques and text mining in energy storage research but fails to provide a comprehensive comparison with existing techniques for interpreting electrochemical signals. This limits the understanding of how the proposed machine learning approach contributes to the field and whether it outperforms or complements existing methods.

Using Machine learning to analyse electrochemical signals in the field of energy storage is totally new. But, utilizing machine-learning on electrochemical signal interpretation in fundamental electrochemistry has been done by the other groups:

- (1) Alan M. Bond's group demonstrated the successful application of ML on voltammetric mechanistic studies to predict the electrochemical mechanism involved in the reaction that express through a cyclic voltammogram. The prediction was done by using simulated duck shaped CV images with various simulated conditions.[6-8]
- (2) Similarly to the work from Cyrille Costentin's group using deep learning to distinguish the mechanism of the electrochemical reaction such as interfacial charge transfers (E step) and/or solution reactions (C steps) from CV based on residual neural network (ResNet) architecture.^[9]

In summary, both of these works are significant advances in the field of analytical electrochemistry. They demonstrate the potential of ML to be used for a variety of tasks in this field, and they could lead to new and improved methods for studying electrochemical reactions. However, the previous methods are focused on analytical or fundamental electrochemistry (which is always found in any electrochemical experiment at higher scan rate). So, our main difference is the study of electrochemical energy storage devices.

We added in the manuscript:

-----Inserted in manuscript at P.5 Line 83-89-----

“Machine learning has been used to predict the electrochemical mechanism involved in the reaction that expresses through a cyclic voltammogram (CV). Deep learning has also been used to distinguish the mechanism of the electrochemical reaction from CV based on residual neural network (ResNet) architecture, and focused on analytical or fundamental electrochemistry. However, the application of machine learning to analyze electrochemical signals in the field of energy storage is still in its early stages.”

-----The content above resides within the manuscript-----

4. Have you conducted any comparative analysis with existing methods to demonstrate the superiority of the capacitive tendency metric? It would be valuable to provide a quantitative comparison and discuss the advantages of your approach over conventional classification techniques in terms of both computational requirements and accuracy.

In conventional way, the method that has been widely used in energy storage field is based on the relationship $i = k_1v + k_2v^{1/2}$, where v is scan rate. However, this method was questioned by Costentin et al.,^[10] on the inappropriate use based on the limitation of this method including the irreversible process during the voltammetry, as well as the complicated capacitive mechanism such as insertion/intercalation in the case of pseudocapacitor which cannot apply this method.

The paper "To Be or Not To Be Pseudocapacitive?" by Brousse et al. [REF 3] is a seminal work in the field of electrochemical capacitors. It is one of the first papers to clearly identify the difference between pseudocapacitive and battery materials. The authors support

their thesis by pointing out that the electrochemical signature of pseudocapacitive materials is very different from that of battery materials. Pseudocapacitive materials typically exhibit rectangular CV curves and triangular GCD curves, while battery materials exhibit rounded CV curves and GCD curves. This difference in electrochemical signature is due to the different mechanisms by which charge is stored in the two types of materials, that the apparent capacitive behavior “anyway if it is pseudo or EDLC” are when:

$$\frac{dQ}{dE} = \text{CONSTANT}, \text{ and the CONSTANT is the capacity.}$$

Here, we compared the method from the previous studies using the popular $v/v^{1/2}$ scan rate diagnosis and the predicted result using machine-learning in our study in **Table R.1**. The results clearly demonstrate that the previous approach is not suitable for peak-shaped CV curves, such as those reported in ref[14, 19] in **Table R1**. Our method suggests that the capacitive tendency in these cases should be a small number.

We added this Table and discussion in supporting manuscript and Table ESI3.

-----Inserted in supporting manuscript at P.37 Line 373-386-----

*“Here, we compared the method from the previous studies using the popular $v/v^{1/2}$ scan rate diagnosis and the predicted result using machine-learning in our study in **Table ESI3**. The results clearly demonstrate the big difference such in the case of ref [14] and ref [19] that the previous approach from literature ($v/v^{1/2}$ scan rate) gave high percentage capacitance, whereas our study suggested the lower percentage of capacitive tendency. Our method suggests that the capacitive tendency in these cases should be a small number since the peak characteristic of CV is dominant. The limitation of the conventional method does not cover some situations as pointed in some references [REF 10 and REF 3]. Unlike the conventional model which relies on a proportionality to scan rate, the present capacitive tendency is a geometric variable that does not indicate whether a dynamic is surface-related or diffusion-related. It is based on an analysis of the signal's shape. It should be noted that the historical concept of pseudocapacitance focuses on this geometric shape more than on a surface vs. diffusion dynamic. Through our approach, we propose to researchers a different indicator, one that is*

more focused on the pure and initial definition.”

-----The content above resides within the manuscript-----

Table R.1. The predicted capacitive tendency compared with the result from the articles.

[Parts of table have been redacted]

Reference	Input CV for the classification	Capacitive contribution with scan rate	Capacitive tendency
[12]		67%	51.90 %
[13]		64%	52.04 %
[14]		74%	38.27 %
[15]		78%	51.94 %
[10]		64%	51.94 %
[16]		n/a	52.10 %
[17]		n/a	51.92 %

[18]		66%	51.90 %
[19]		70%	18.26 %
[20]		63%	96.03 %
[21]		n/a	96.10 %
[22]		93%	95.80 %
[23]		66%	63.45 %
[24]		93%	51.93 %
[25]		n/a	51.90 %
[26]		n/a	51.97 %
[27]		80%	51.98 %

[28]		n/a	3.80 %
[29]		75%	95.47 %
[30]		n/a	96.20 %

Minor issues:

- The article lacks sufficient contextualization within the broader field of energy storage research. While the article discusses the significance of distinguishing between capacitive and battery-type signals, it does not sufficiently connect this research to the current state of the field or highlight how the findings contribute to advancing energy storage technologies. For instance, the use of machine learning approach to analyse the impact of physicochemical features of carbon electrodes on the capacitive performance of supercapacitors could be mentioned.

We added further discussion on this point in the manuscript.

-----Inserted in manuscript at P.15 Line 294-299-----

*“The understanding of the origin of the electrochemical behavior is the key point for the deep knowledge and for the future development of the electrode materials. TB robots have been used to study the physicochemical features of a variety of electrode materials, including carbon electrodes, MOFs, COFs, graphite, NMC, and MXenes materials, determining the capacitive tendency of the CV in ‘continuum region from the recent papers (as shown in **Figure ESI 21**).”*

-----The content above resides within the manuscript-----

- In the article, you mentioned that there can be an overlap between battery and pseudocapacitor signals due to their faradaic nature. It would be beneficial to discuss some examples or case studies where this overlap occurs and how the proposed capacitive tendency metric effectively distinguishes between them. This would provide

concrete evidence of the robustness and accuracy of your approach.

The case study of the overlap of electrochemical signal characteristics referred to the paper of the proposed continuum transition between supercapacitor and battery CV characteristics (Nature Energy, Fleischmann et al.^[11]).

Overlap region

[redacted]

From the supplementally information in Figure ESI 21aII, and 21bII as shown below, we transform the concrete analysis of the overlap region that is where capacitive behavior is found. Our approach can give a quantitative predicted result with exact percentage of the capacitive tendency of any CV in the transition region.

Figure ESI 21 | The predicted representative electrochemical signals (from Nature Energy, Fleischmann et al.^[11]): a-I) porous carbon with solvated ion adsorption, a-II) porous carbon with partially solvated ion adsorption, a-III) graphite with desolvated Li⁺ intercalation, b-I) MXene with hydrated Li⁺ adsorption, b-II) MXene with partially desolvated Li⁺ intercalation, b-III) layered LiNi_{1/3}Mn_{1/3}Co_{1/3}O₂, NMC with desolvated Li⁺ intercalation, and c) % capacitive tendency.

We added in the manuscript:

-----Inserted in manuscript at P.15 Line 295-299-----

*“The trained model has been used to study the physicochemical features of a variety of electrode materials, including carbon electrodes, MOFs, COFs, graphite, NMC, and MXenes materials, determining the capacitive tendency of the CV in ‘continuum region from the recent papers (as shown in **Figure ESI 21**).*”

-----The content above resides within the manuscript-----

7. Could you please elaborate on the dataset used for training the supervised machine learning algorithm? Specifically, how diverse is the dataset in terms of electrode materials and electrochemical behaviours? It would be helpful to understand the representativeness of the dataset and its impact on the performance and generalizability of the proposed classification approach.

The logic behind our classification is only based on binary image identification. Since we aimed to distinguish the electrochemical signals (CV and GCD) between battery type vs pseudocapacitor type, we trained the ML model with CV/GCD of these type of electrode materials (especially the non-ambiguous ones). The study only focuses on the identification of the electrochemical signal shapes and shape variation such as the box vs peak shaped CV, and the triangular vs plateau shaped GCD as shown in **Figure R5** below.

[redacted]

Where all images of CV and GCD training datasets were published in Github repository (https://github.com/ice555mee/TB-robot_code-data):

- Battery CV training data: https://github.com/ice555mee/TB-robot_code-data/tree/main/CV%20classification/CV%20python/CV%20classification%20process%203/Battery%20training%20data
- Pseudocapacitor CV training data: https://github.com/ice555mee/TB-robot_code-data/tree/main/CV%20classification/CV%20python/CV%20classification%20process%203/Pseudocapacitor%20training%20data
- Battery GCD training data: https://github.com/ice555mee/TB-robot_code-data/tree/main/GCD%20classification/GCD%20Py/Battery%20training%20data
- Pseudocapacitor GCD training data: https://github.com/ice555mee/TB-robot_code-data/tree/main/GCD%20classification/GCD%20Py/Pseudocapacitor%20training%20data

We added in the manuscript:

-----Inserted in manuscript at P.7 Line 125-128-----

*These training processes is based on binary classification of electrochemical signal, such as the box vs peak shaped CV, and the triangular vs plateau shaped GCD, as represented in **Figure ESI 6**, where all image datasets used are available on Github.*

-----The content above resides within the manuscript-----

8. The article lacks clarity in explaining the methodology used for dataset construction and the specific processes involved in the machine learning classification. Important details such as the selection criteria for training and validation datasets, data preprocessing techniques, and hyperparameter tuning are not sufficiently explained. This hinders reproducibility and makes it challenging for readers to evaluate the methodology. Moreover, in the Methods section, all Processes and Outputs shall be described in chronological order.

We explained more details on the methodology of the classification. For data collections, PyMuPDF and OpenCV were used to extract the figures from the articles and only CV and GCD were collected and put into categories (1. Box shaped CV, 2. Peak shaped CV, 3.

Triangular GCD, and 4. Plateau GCD), which were then used as the training data. The ambiguous shaped CV and GCD is put in the categories of testing data. In the data processing step, we simply collected them without any processing in order to clean the image of CV or GCD such as removing the axis or separating the curve according to the different scan rates or labels. This purpose is that we would like to finally create a tool for any users to just simply input their data (image of CV or GCD) directly without any pre-processing for quick and easy analysis.

8.1 Data collection and model training

We highlighted in the manuscript:

-----Inserted in manuscript at P.7 Line 117-121-----

“In the present paper, all datasets are in the form of images extracted using PyMuPDF library in Python language from more than 3,300 scientific papers. The first dataset, or Output 1, was obtained by figures extracting using OpenCV which provides (2,979) GCD, (5,598) CV and other images such as crystal structure image (which will not be used in the further classification steps).”

-----The content above resides within the manuscript-----

We added in the supporting manuscript:

-----Inserted in supporting manuscript at P.6 Line 111-115-----

3.Data collection, model training, and classification

*The first step was to extract the figures from the scientific paper using the PyMuPDF library in Python. Each figure could contain multiple CVs or GCDs. The OpenCV library was then used to separate each CV or GCD image. The resulting dataset contained CVs, GCDs, and other images, such as the author's image, the journal's logo, and illustrations. This is illustrated in **Figure ESI 4**.*

[redacted]

*Since the number of images extracted from the articles was large, we needed to screen and collect only the CVs and GCDs from the entire image dataset. To do this, we quickly classified the extracted images using the ResNet50 model, which only collects CV and GCD images (this step is called Process 1). This is illustrated in **Figure ESI 5**. Firstly, the images of CV and GCD were manually labelled by human to be used as the training dataset for Process 1 (distinguishing CV/GCD from the other images). Process1 was then performed (using ResNet50 architecture) to classify and collect only CV and GCD images from all unclassified images. The prediction (Output1) will finally compose of 3 categories:*

- 1. CV*
- 2. GCD*
- 3. Other images.*

Here, only the CV and GCD were then used for the classification in the next step (Process2, 3, 4, and 5) later.

[redacted]

-----The content above resides within the manuscript-----

We edited in the manuscript:

-----Inserted in manuscript at P.7 Line 121-125-----

In the training process of GCD (process 2) and CV(process 3) classification, CV and GCD images were firstly labeled as belonging to one of two classes, namely battery or pseudocapacitor following the criteria of non-ambiguous signal shape (which can be put into four categories: 1. Box shaped CV, 2. Peak shaped CV, 3. Triangular GCD, and 4. Plateau GCD) for 80 % of total data, where 20 % of total data was used as testing data.

-----The content above resides within the manuscript-----

8.2 Labeling of GCDs and CVs for process 2,3

We added in the manuscript:

-----Inserted in manuscript at P.7 Line 126-128-----

*“, such as the box vs peak shaped CV, and the triangular vs plateau shaped GCD, as represented in **Figure ESI 6**, where all image datasets used are available on Github.”*

-----The content above resides within the manuscript-----

And,

-----Inserted in supporting manuscript at P.11-----

***Figure ESI 6** | The representative of training datasets of CV (a) box, (b) peak characteristic, and GCD (c) triangular, and (d) plateau characteristic.*

-----The content above resides within the manuscript-----

8.3 Labeling of GCDs and CVs for process 4,5

We highlighted in the manuscript:

-----Inserted in manuscript at P.7 Line 129-131-----

*“From Process 3, Output 3 was obtained and categorized into three types of training sets: 100 % battery, 50 % battery/pseudocapacitor, and 100 % pseudocapacitor. This output was then further refined in Processes 4 and 5, as illustrated in **Figure 3b**.”*

-----The content above resides within the manuscript-----

8.4 Classification architecture

Hence, the criteria of choosing the classification architecture are to use the deep layers of ML convolution layers with complex structure to be able to detect those generic informative details such as frame, axis, multiple line, colors, fonts, etc. as shown in **Figure R6**. So, we chose to test the different architectures as described in the manuscript (Validation of the classification architectures). As shown in Table ESI2, the classification architectures were evaluated and validated according to Accuracy, Sensitivity, Specificity, Precision, and F1-score.

[redacted]

Table ESI 2 | *GCD* and *CV* classification comparison based on evaluation values obtained from five different architectures; *ResNet50*, *MobileNetV2*, *VGG16*, *Xception*, and *8-Layer CNN*.

CNN-Model	Accuracy (%)	Sensitivity (%)	Specificity (%)	Precision (%)	F1 -Score
GCD					
ResNet50	94.22	93.84	94.45	94.16	93.99
MobileNetV2	93.11	92.56	93.07	93.12	92.82
VGG16	92.22	92.24	94.61	91.78	91.99
Xception	93.77	93.98	96.49	93.33	93.60
8-Layer CNN	94.00	93.73	94.77	93.82	93.78
CV					
ResNet50	95.80	93.52	96.74	94.65	94.07
MobileNetV2	94.64	92.62	96.42	93.24	92.92
VGG16	94.36	93.12	97.13	91.53	92.28
Xception	93.04	88.87	94.35	91.31	90.00
8-Layer CNN	93.65	89.08	94.26	92.77	90.74

9. How *GCD* and *CV* pictures are extracted from articles and recognized among other Figures? Did you check that the adopted measure units are the same in these graphs? How did you train Process 1? How did you label the *GCD* and *CV* pictures so that they belong to one of these two classes? How did you label the Output 3 into the three types of training sets for Process 3 and 4?

How *GCD* and *CV* pictures are extracted from articles and recognized among other Figures?

In the first step, the figures in the article were extracted using PyMuPDF library in Python language from scientific paper. In one figure could be composed of many *CV*s or *GCD*s. Here, OpenCV was used to separate each *CV* or *GCD* images. From this step, the overall dataset will contain *CV*, *GCD*, and other images such as the image of the author, the logo of the journal, the illustration, etc. as illustrated in **Figure EIS 4**.

Since the number of images that could be extracted from the articles are large and we need to screen and collect *CV* and *GCD* images from the whole image dataset. Hence, these extracted images were quickly classified to collect only *CV* and *GCD* images using ResNet50 model (this

step called Process 1), as illustrated in **Figure ESI 5**.

Process 1:

Firstly, the images of CV and GCD were manually labelled by human to be used as the training dataset for Process 1 (distinguishing CV/GCD from the other images). Process1 was then performed (using ResNet50 architecture) to classify and collect only CV and GCD images from all unclassified images. The prediction (Output1) will finally compose of 3 categories:

4. CV
5. GCD
6. Other images.

Here, only the CV and GCD were then used for the classification in the next step (Process2, 3, 4, and 5) later.

We added in the supporting manuscript:

“3.Data collection, model training, and classification

*The first step was to extract the figures from the scientific paper using the PyMuPDF library in Python. Each figure could contain multiple CVs or GCDs. The OpenCV library was then used to separate each CV or GCD image. The resulting dataset contained CVs, GCDs, and other images, such as the author's image, the journal's logo, and illustrations. This is illustrated in **Figure ESI 4**.*

[redacted]

*Since the number of images extracted from the articles was large, we needed to screen and collect only the CVs and GCDs from the entire image dataset. To do this, we quickly classified the extracted images using the ResNet50 model, which only collects CV and GCD images (this step is called Process 1). This is illustrated in **Figure ESI 5**. Firstly, the images of CV and GCD were manually labelled by human to be used as the training dataset for Process 1 (distinguishing CV/GCD from the other images). Process1 was then performed (using ResNet50 architecture) to classify and collect only CV and GCD images from all unclassified images. The prediction (Output1) will finally compose of 3 categories:*

- 7. CV*
- 8. GCD*
- 9. Other images.*

Here, only the CV and GCD were then used for the classification in the next step (Process2, 3, 4, and 5) later.

[redacted]

-----Inserted in supporting manuscript at P.6 Line 111-115-----

5.2 Did you check that the adopted measure units are the same in these graphs?

We only took consideration on the shape of CV or GCD, so we did not do any value

measurement or checking the unit from the figures, since the large number of figures has different axis value and different unit.

How did you train Process 1?

As aforementioned in the answer of Question 5.1, in *Figure ESI 5*, the images of CV and GCD were manually labelled by human to be used as the training dataset for Process 1 (distinguishing CV/GCD from the other images). Process1 was then performed (using ResNet50 architecture) to classify and collect only CV and GCD images from all unclassified images. The prediction (Output1) will finally compose of 3 categories:

10. CV
11. GCD
12. Other images.

How did you label the GCD and CV pictures so that they belong to one of these two classes?

The images of CV and GCD were manually labelled by human that the GCD images normally look like the triangular shape or plateau shape, where CV images normally look like box shape or peak or duck shapes as shown in **Figure R5**.

[redacted]

How did you label the Output 3 into the three types of training sets for Process 3 and 4?

We edited the new version of Figure 3, as shown below. The output 3 from Process 3 was then further selectively labeled as 4 categories as 100 % battery, 50 % battery/pseudocapacitor, and 100 % pseudocapacitor, according to the confident percentage from the prediction from Process 3, as illustrated in **Figure 3b**.

Figure 3 | (a) CV and GCD datasets obtained after classification by Process 1, splitting them into training and testing datasets for further GCD and CV classification in Process 2 and Process 3, respectively. (b) The outputs from Process 3 are used in this final classification step to obtain the capacitive tendency based on percentage confidence rating of the prediction. (c) Table of processes, inputs and outputs performed/used to obtain these results.

9. “Moreover, cross-validation was performed with the experts in the field with the number of meetings”: what do you mean? Which types of cross-validations were performed? Who were the experts? Please describe such cross-validation with more quantitative arguments.

The cross-validation was done by using improved training datasets (or using the different training datasets collected by different experts) to optimize the performance of the classification each time.

We edited in the manuscript:

-----Inserted in manuscript at P.7 Line 135-136-----

“Moreover, cross-validation was performed with the experts in the field to generate the different training datasets for the optimizing of the classification performance.”

-----The content above resides within the manuscript-----

10. Please better describe why two models (Process 4 and 5) have been trained to predict the capacitive tendency if the output figure of merit is just one. In this sense, Output 4 and 5 seem redundant (they should be complementary with each other).

Basically, the training datasets for process 4 and 5 in CV classification were the output from classification in process 3 that the classes of output are categorized by % confidence: (Class I) ~0% confidence, (Class II) ~50% confidence, (Class III) ~100% confidence as pseudocapacitor type as illustrated below:

Figure R7. CV images classified in process 3 giving three classes of output.

We highlighted in the manuscript:

-----Inserted in manuscript at P.7 Line 129-131-----

*“From Process 3, Output 3 was obtained and categorized into three types of training sets: 100 % battery, 50 % battery/pseudocapacitor, and 100 % pseudocapacitor. This output was then further refined in Processes 4 and 5, as illustrated in **Figure 3b.**”*

-----The content above resides within the manuscript-----

11. The subsection “The issues surrounding electrochemical signal identification” appears as a repetition of Introduction rather than a Results. Please improve the readability of the article by removing redundant parts.

We removed this part and integrated it to improve the introduction part.

-----Inserted in manuscript at P.3 Line 48-52 -----

“However, some faradaic electrode materials including pseudocapacitors display electrochemical signals similar to those of EDLCs, such as the rectangular/quasi-rectangular CV and the sloping GCD curves,^[6, 7] found in a variety of transition metal oxides (RuO₂,^[8] MnO₂^[9, 10]), conducting polymers (poly(3,4-ethylenedioxythiophene)^[11, 12], polyaniline^[13, 14]), and carbides (MXene)^[15].”

-----The content above resides within the manuscript-----

And,

-----Inserted in manuscript at P.3 Line 56-58-----

*“Indeed, electrochemical signals are numerous and complex, varying according to the choice of electrode materials, as shown in **Figure 1**, hence the difficulty in identifying and categorizing these materials based on electrochemical signals.”*

-----The content above resides within the manuscript-----

12. In the analysis carried out in Figure 7, were the considered articles outside training set?

Yes, the analysis was performed by using a number of articles that contain keyword ‘battery’ or ‘pseudocapacitor’ which is outside the training dataset. This method will only compare the results from the image classification of CV and GCD by ML with the type of electrode material defined by the authors.

We added in the manuscript:

-----Inserted in manuscript at P.16 Line 305-306-----

“(used articles outside the training dataset)”

-----The content above resides within the manuscript-----

13. The article briefly mentions the limitation of electrochemical signals deviating from ideal curves, but it does not extensively discuss other potential limitations of the proposed machine learning approach. Furthermore, the article does not provide a detailed discussion on future directions for improving the methodology or addressing these limitations.

The potential drawback of this ML approach can be the lack of data mining of the image data such as the plot label, scan rate, or electrolyte. This requires a huge work and human power to integrate image recognition and text mining out of the image data. The addition of these improvements to overcome these limitations would benefit and give an ultimate tool for this analysis of electrochemical signals.

We discussed more of the limitations and the improvements in the conclusion part.

-----Inserted in manuscript at P.19 Line 360-364-----

“However, a potential drawback of the current classifier is that it can only predict the resistive tendency of electrochemical signals based on CV/GCD image data. A more comprehensive classifier by featuring text-mining of material information of a hidden information such as labels, scan rate, electrolytes in the figure could be an ultimate strategy for future perspectives on artificial intelligence for energy storage technology.”

-----The content above resides within the manuscript-----

14. Please improve English language and correct typos (e.g., caption of Fig. 6, table headings of Fig. 6).

We have corrected the typos at Figure 6.

Reference

1. Huang, S. and J.M. Cole, *BatteryDataExtractor: battery-aware text-mining software embedded with BERT models*. Chem Sci, 2022. **13**(39): p. 11487-11495.
2. El-Bousiydy, H., et al., *What Can Text Mining Tell Us About Lithium-Ion Battery Researchers' Habits?* Batteries & Supercaps, 2021. **4**(5): p. 758-766.
3. Mahbub, R., et al., *Text mining for processing conditions of solid-state battery electrolytes*. Electrochemistry Communications, 2020. **121**.
4. El-Bousiydy, H., et al., *LIBAC: An Annotated Corpus for Automated "Reading" of the Lithium-Ion Battery Research Literature*. Chemistry of Materials, 2023. **35**(5): p. 1849-1857.
5. Chen, H., E. Kätelhön, and R.G. Compton, *Machine learning in fundamental electrochemistry: Recent advances and future opportunities*. Current Opinion in Electrochemistry, 2023. **38**: p. 101214.
6. Bond, A.M., et al., *Opportunities and challenges in applying machine learning to voltammetric mechanistic studies*. Current Opinion in Electrochemistry, 2022. **34**: p. 101009.
7. Gundry, L., et al., *Inclusion of multiple cycling of potential in the deep neural network classification of voltammetric reaction mechanisms*. Faraday Discussions, 2022. **233**(0): p. 44-57.
8. Kennedy, G.F., J. Zhang, and A.M. Bond, *Automatically Identifying Electrode Reaction Mechanisms Using Deep Neural Networks*. Analytical Chemistry, 2019. **91**(19): p. 12220-12227.
9. Hoar, B.B., et al., *Electrochemical Mechanistic Analysis from Cyclic Voltammograms Based on Deep Learning*. ACS Measurement Science Au, 2022. **2**(6): p. 595-604.
10. Costentin, C., *Electrochemical Energy Storage: Questioning the Popular $v/v_{1/2}$ Scan Rate Diagnosis in Cyclic Voltammetry*. The Journal of Physical Chemistry Letters, 2020. **11**(22): p. 9846-9849.
11. Fleischmann, S., et al., *Continuous transition from double-layer to Faradaic charge storage in confined electrolytes*. Nature Energy, 2022. **7**(3): p. 222-228.
12. Shao, H., et al., *Electrochemical study of pseudocapacitive behavior of Ti_3C_2Tx MXene material in aqueous electrolytes*. Energy Storage Materials, 2019. **18**: p. 456-461.
13. Hu, L., et al. *Cu_2Se Nanoparticles Encapsulated by Nitrogen-Doped Carbon Nanofibers for Efficient Sodium Storage*. Nanomaterials, 2020. **10**, DOI: 10.3390/nano10020302.
14. Jiang, Y. and J. Liu, *Definitions of Pseudocapacitive Materials: A Brief Review*. ENERGY & ENVIRONMENTAL MATERIALS, 2019. **2**(1): p. 30-37.
15. Hu, L. and C. Shang *$Co_3V_2O_8$ Nanoparticles Supported on Reduced Graphene Oxide for Efficient Lithium Storage*. Nanomaterials, 2020. **10**, DOI: 10.3390/nano10040740.
16. Zhang, J., et al., *Urchin-Like Fe_3Se_4 Hierarchitectures: A Novel Pseudocapacitive Sodium-Ion Storage Anode with Prominent Rate and Cycling Properties*. Small, 2020. **16**(26): p. 2000504.
17. Mishra, N.K., R. Mondal, and P. Singh, *Synthesis, characterizations and electrochemical performances of anhydrous CoC_2O_4 nanorods for pseudocapacitive energy storage applications*. RSC Advances, 2021. **11**(54): p. 33926-33937.
18. Liu, X., et al., *Ultrafine MoO_3 nanoparticles embedded in porous carbon nanofibers as anodes for high-performance lithium-ion batteries*. Materials Chemistry Frontiers, 2019. **3**(1): p. 120-126.
19. Chong, S., et al., *Potassium Nickel Iron Hexacyanoferrate as Ultra-Long-Life Cathode Material for Potassium-Ion Batteries with High Energy Density*. ACS Nano, 2020. **14**(8): p. 9807-9818.
20. Wang, G., et al., *Hierarchical Carbon Nanosheet Assembly with SiO_x Incorporation*

- and Nitrogen Doping Achieves Enhanced Lithium Ion Storage Performance*. *Advanced Energy and Sustainability Research*, 2021. **2**(7): p. 2100026.
21. Zhang, W., et al., *Mesoporous TiO₂/TiC@C Composite Membranes with Stable TiO₂-C Interface for Robust Lithium Storage*. *iScience*, 2018. **3**: p. 149-160.
 22. Chen, H., et al., *A new spinel high-entropy oxide (Mg_{0.2}Ti_{0.2}Zn_{0.2}Cu_{0.2}Fe_{0.2})₃O₄ with fast reaction kinetics and excellent stability as an anode material for lithium ion batteries*. *RSC Advances*, 2020. **10**(16): p. 9736-9744.
 23. Zhang, C., et al., *Polyimide@Ketjenblack Composite: A Porous Organic Cathode for Fast Rechargeable Potassium-Ion Batteries*. *Small*, 2020. **16**(38): p. 2002953.
 24. Xu, W., et al., *Sn nanocrystals embedded in porous TiO₂/C with improved capacity for sodium-ion batteries*. *Inorganic Chemistry Frontiers*, 2019. **6**(10): p. 2675-2681.
 25. Wei, T., et al., *An electrochemically induced bilayered structure facilitates long-life zinc storage of vanadium dioxide*. *Journal of Materials Chemistry A*, 2018. **6**(17): p. 8006-8012.
 26. Li, H., et al., *A High-Performance Sodium-Ion Hybrid Capacitor Constructed by Metal–Organic Framework–Derived Anode and Cathode Materials*. *Advanced Functional Materials*, 2018. **28**(30): p. 1800757.
 27. Li, S., et al., *Encapsulation of MnS Nanocrystals into N, S-Co-doped Carbon as Anode Material for Full Cell Sodium-Ion Capacitors*. *Nano-Micro Letters*, 2020. **12**(1): p. 34.
 28. Ren, C., et al., *Hierarchical Porous Integrated Co_{1-x}S/CoFe₂O₄@rGO Nanoflowers Fabricated via Temperature-Controlled In Situ Calcining Sulfurization of Multivariate CoFe-MOF-74@rGO for High-Performance Supercapacitor*. *Advanced Functional Materials*, 2020. **30**(45): p. 2004519.
 29. Jia, H., et al., *Advanced ZnSnS₃@rGO Anode Material for Superior Sodium-Ion and Lithium-Ion Storage with Ultralong Cycle Life*. *ChemElectroChem*, 2019. **6**(4): p. 1183-1191.
 30. Gong, Y., et al., *Electric Double-layer Capacitance and Pseudocapacitance Contributions to the Oxidative Modification of Helical Carbon Nanofibers*. *International Journal of Electrochemical Science*, 2020. **15**(8): p. 7508-7519.
 31. Goikolea, E., et al., *Synthesis of nanosized MnO₂ prepared by the polyol method and its application in high power supercapacitors*. *Materials for Renewable and Sustainable Energy*, 2013. **2**(3): p. 16.
 32. Zúcalová, M., et al., *LiNi_{1/3}Mn_{1/3}Co_{1/3}O₂ with morphology optimized for novel concept of 3D Li accumulator*. *International Journal of Energy Research*, 2020. **44**(11): p. 9082-9092.
 33. Le Calvez, E., et al., *Investigating the Perovskite Ag_{1-3x}La_xNbO₃ as a High-Rate Negative Electrode for Li-Ion Batteries*. *Frontiers in Chemistry*, 2022. **10**.
 34. Miranda, J., et al., *Revisiting Rb₂TiNb₆O₁₈ as electrode materials for energy storage devices*. *Electrochemistry Communications*, 2022. **137**: p. 107249.

REVIEWERS' COMMENTS

Reviewer #1 (Remarks to the Author):

Reviewer's general comment: The work focused on a machine learning method with the capacitive tendency for classifying battery and pseudocapacitor materials. The manuscript is within the scope of the Journal. To help improve the paper's quality, my suggestions and comments are shown below.

1) There is one article online in research square: A Novel Approach for Classifying Battery and Pseudocapacitor Materials Using Capacitive Tendency and Supervised Machine Learning. The content is almost similar. Please ensure there is no conflict of interest for publication.

(2) The author mention: 'Our work is the first to use supervised machine learning to interpret electrochemical signal shape, specifically CV and GCD images'. After checking in google scholar, the reviewer found following articles:

K Khosravinia, A Kiani. Unlocking pseudocapacitors prolonged electrode fabrication via ultra-short laser pulses and machine learning

Wang, T., Pan, R., Martins, M.L. et al. Machine-learning-assisted material discovery of oxygen-rich highly porous carbon active materials for aqueous supercapacitors. Nat Commun 14, 4607 (2023).

P Puthongkham, S Wirojsaengthong, A Suea-Ngam. Machine learning and chemometrics for electrochemical sensors: moving forward to the future of analytical chemistry. Analyst, 2021, 146, 6351-6364

The authors need to justify the originality with these works.

(3) As mentioned by authors, 'The originality of this work lies in its use of machine learning (ML) to quickly and accurately interpret electrochemical signal images and transform them into accurate values. This is made possible by the large database of electrochemical energy storage images that is available to the ML model'. Actually, machine learning (ML) to quickly and accurately interpret images has been widely used in cancer detection. The main difference and breakthrough of ML in electrochemical signal images between cancer detection is better to be provided, so as to justify the contribution of the work.

(4) The manuscript needs to be carefully checked to avoid grammar errors.

(5) Comment 7: Compare your approaches used in your study to the others in terms of their advantages and drawbacks. It is better to add the detailed comparison in the main context with references.

(6) Regarding the generality and universality of the proposed approach, breakthrough out of the training database boundary should be provided, for performance prediction, material classification and etc. Is it possible for authors to add relevant contents on this?

Reviewer #3 (Remarks to the Author):

The Authors have replied to all raised issues in a convincing way and improved the manuscript accordingly.

Reviewer 1

Reviewer's general comment: The work focused on a machine learning method with the capacitive tendency for classifying battery and pseudocapacitor materials. The manuscript is within the scope of the Journal. To help improve the paper's quality, my suggestions and comments are shown below.

We understand the intentions of reviewer 1 and thank him for this analysis, which enhances the wide audience of Nature communications.

1. There is one article online in research square: A Novel Approach for Classifying Battery and Pseudocapacitor Materials Using Capacitive Tendency and Supervised Machine Learning. The content is almost similar. Please ensure there is no conflict of interest for publication.

There is no conflict of interest, The article in Research Square corresponds to our work, and when we submitted it to the Nature Comm journal, we were offered the possibility of pre-depositing it in Research Square.

2. The author mention: ' Our work is the first to use supervised machine learning to interpret electrochemical signal shape, specifically CV and GCD images ' . After checking in google scholar, the reviewer found following articles:

K Khosravinia, A Kiani. Unlocking pseudocapacitors prolonged electrode fabrication via ultra-short laser pulses and machine learning
Wang, T., Pan, R., Martins, M.L. et al. Machine-learning-assisted material discovery of oxygen-rich highly porous carbon active materials for aqueous supercapacitors. Nat Commun 14, 4607 (2023).

In the present scientific article, the author uses machine learning to select the best precursor to predict the specific capacitance.

P Puthongkham, S Wirojsaengthong, A Suea-Ngam. Machine learning and chemometrics for electrochemical sensors: moving forward to the future of analytical chemistry. Analyst, 2021, 146, 6351-6364.

This paper is a minireview summarizing recent applications of machine learning and experimental designs in electroanalytical chemistry.

The authors need to justify the originality with these works.

We thank the reviewer for this information. We have added this state of the art to the main manuscript. However, we would like to draw the reviewer's attention to the fact that these articles focus on the correlation between materials and properties. We emphasize our innovation: the analysis of electrochemical signal shape.

3. As mentioned by authors, 'The originality of this work lies in its use of machine learning (ML) to quickly and accurately interpret electrochemical signal images and transform them into accurate values. This is made possible by the large database of electrochemical energy storage images that is available to the ML model'. Actually, machine learning (ML) to quickly and accurately interpret images has been widely used in cancer detection. The main difference and breakthrough of ML in electrochemical signal images between cancer detection is better to be provided, so as to justify the contribution of the work.

The basic concept of image recognition is the same, whatever the technology: transforming an image into pixels and discovering a pattern through learning. The difference between all these types of learning is the particularity of the images to be analyzed. This is what will decide which neural network to use. In the manuscript, we don't think it's appropriate to compare our approach specifically to cancer detection. However, we have added a more general text on image recognition:

Image recognition is used in many fields, such as facial recognition, cancer detection and autonomous cars. All these models have been trained using a supervised or semi-supervised deep learning approach, in order to teach the model, the pattern best suited to the situation. The difference between the techniques lies in the choice of neural network, which must be adapted to the specific problem. In our case, the main difficulty was to differentiate the figures representing a CV and a GCD from the other graphs.

4. The manuscript needs to be carefully checked to avoid grammar errors.

We checked the grammar and found no more issues.

5. Comment 7: Compare your approaches used in your study to the others in terms of their advantages and drawbacks. It is better to add the detailed comparison in the main context with references.

The capacitive tendency represents the shape of the electrochemical signal, this variable is new and different of other previous analysis, mainly the surface and diffusional contribution analysis.

In the present case, the Capacitive tendency is useful if the reader/ scientist wants to compare different materials and to check the general trend of the electrochemical signal not to extract the surface contribution to the diffusional contribution.

In the main manuscript, we added this sentence:

In comparison to other studies, the capacitive tendency analyses the shape of the electrochemical signal. Unfortunately, the capacitive tendency doesn't provide the surface contribution or the diffusional contribution inside the cyclic voltammetry.

6. Regarding the generality and universality of the proposed approach, breakthrough out of the training database boundary should be provided, for performance prediction, material classification and etc. Is it possible for authors to add relevant contents on this?

We added the following text in the main manuscript:

The training database boundary is fixed using only scientific data with the pseudocapacitor or battery keywords associated. It is recommended for reader to use the present model to compare signal associated to EDLC, pseudocapacitor and Metal-ion battery. The present model isn't adapted to redox flow battery, and fuel-cells. Moreover, the present model doesn't provide any performance predictions. The typical useful application is to compare the same family of materials (*i.e.*, MOF, NMC, MXene) but presenting a different electrochemical behavior. That is the generality and universality of this study.